psychology

cooperation, competition, preschoolers, intergroup behaviour, minimal groups, social inclusion

**Author for correspondence:**
Theo Toppe
e-mail: theo_toppe@eva.mpg.de

# The influence of cooperation and competition on preschoolers' prosociality toward in-group and out-group members

Theo Toppe[1], Susanne Hardecker[2], Franca Zerres[3] and Daniel B. M. Haun[1,4]

[1]Department of Comparative Cultural Psychology, Max Planck Institute for Evolutionary Anthropology, Leipzig, Germany
[2]SRH University of Applied Health Sciences Gera, Gera, Germany
[3]Department of Early Child Development and Culture, and [4]Leipzig Research Center for Early Child Development, Faculty of Education, Leipzig University, Leipzig, Germany

(iD) TT, 0000-0002-4241-2092

Past research suggests that children favour their in-group members over out-group members as indicated by selective prosociality such as sharing or social inclusion. This preregistered study examined how playing a cooperative, competitive or solitary game influences German 4- to 6-year-olds' in-group bias and their general willingness to act prosocially, independent of the recipient's group membership ($N = 144$). After playing the game, experimenters introduced minimal groups and assessed children's sharing with an in-group and an out-group member as well as their social inclusion of an out-group member into an in-group interaction. Furthermore, we assessed children's physical engagement and parents' social dominance orientation (SDO)—a scale indicating the preference for inequality among social groups—to learn more about inter-individual differences in children's prosocial behaviours. Results suggest that children showed a stronger physical engagement while playing competitively as compared with cooperatively or alone. The different gaming contexts did not impact children's subsequent in-group bias or general willingness to act prosocially. Parental SDO was not linked to children's prosocial behaviours. These results indicate that competition can immediately affect children's behaviour while playing but raise doubt on the importance of cooperative and competitive play for children's subsequent intergroup and prosocial behaviour.

# 1. Introduction

Humans interact with each other constantly. Humans' interactions are transient and vary across situations. Cooperation and competition are two complementary forms of our interactions and are assumed to have a significant impact on humans' social behaviour. The current study investigated how cooperation and competition influence children's intergroup behaviour and prosociality.

## 1.1. Social interdependence theory

Social interdependence theory states that cooperation and competition are the two fundamental forms of interaction [1,2]. When cooperating, interaction partners have a common goal and win and lose together; when competing, interaction partners have opposite goals, and one person's win is tied to the other person's loss. Individuals' goals can also be independent. This is the case when there is no relation between the achievement of individuals' goals.

According to the social interdependence theory, these three different forms of social interdependence elicit different psychological orientations influencing our social behaviour [1,2]. A cooperative orientation comprises anticipated prosociality of others, an increased prosociality toward these and an egalitarian morality [3]. An opposite effect is assumed by competitive contexts (i.e. when having opposite goals) that elicit a competitive orientation characterized by anticipated resistance and decreased prosociality [4]. Neutral contexts, in which agents' goals are independent, do not change the expectation of others' prosociality [5]. The respective orientation not only changes prosocial behaviours within the context of their occurrence but is also transferred to new situations [6]. Such spill-over effects of cooperation and competition have been found for children's intergroup behaviour and general prosociality.

## 1.2. Children's intergroup behaviour

Social identity theory [7,8] assumes that individuals favour their own group over an out-group. This in-group bias results from two processes: first, individuals tend to categorize others and themselves by meaningful differences creating a 'fundamental lens' through which they see others and themselves [9, p. 2] and which serves as a basis for their self-concept. Second, individuals strive for high self-esteem and tend to enhance themselves. That is, one has the desire to be positively distinct from others. These two processes lead individuals to evaluate their entire in-group more favourably than an out-group, which indirectly maintains high self-esteem.

Past research suggests that these processes emerge in early development. From infant age onwards, humans categorize others based on markers, such as gender or ethnicity [10,11], and favour in-group members of such groups [12,13]. Around preschool age, children begin to show an in-group bias in contexts of groups that have been established by arbitrary criteria (e.g. randomly selected colours of clothing items). Such 'minimal' groups are particularly interesting since they indicate how mere group membership impacts human behaviour without the interference of confounders, such as familiarity [14]. Besides their expectation of minimal in-group members to support each other [15,16], 3- to 5-year-old children show more liking of in-group members and share more resources with them than with out-group members [17–21]. Also, preschoolers are less likely to include out-group members into an in-group interaction as compared with a control context without groups [22]. However, evidence on the developmental trajectory of children's in-group bias in the context of minimal groups is still mixed. While some studies found an in-group bias in minimal group situations to emerge at age 3 [19,22,23], other investigations suggested an onset from around age 5 to 6 [17,24–26]. Notably, some studies did not find any in-group bias for sharing behaviour throughout preschool age (e.g. [24]).

## 1.3. The effect of cooperation and competition on children's in-group bias

Prior work assumes that in-group bias lays the foundation for prejudice and intergroup conflicts [27]. In the enterprise to reduce in-group bias, the elicitation of a cooperative psychological orientation has been a promising approach. The underlying idea is that individuals perceive the relation between groups as cooperative when having a cooperative orientation, which reduces (or even eliminates) their in-group bias. By contrast, competitive orientations pronounce in-group bias.

Previous evidence resonates with these predictions. For example, when facing intergroup competition, preschoolers shared and cooperated at higher rates with their in-group members as

opposed to out-group members [28,29]. Further, 5- to 10-year-olds reported lower prosocial intentions toward out-group members in a competitive compared with a non-competitive scenario [30]. In an interesting study, Spielman [31] primed 6-year-old children with stories either including a competitive or neutral interaction of peers and thereby elicited a competitive versus a neutral (i.e. non-competitive) psychological orientation. The competitive story was about two children having a race, while the neutral story was about two children playing together on a playground. Hereafter, children distributed resources with in-group and out-group members. Children showed a stronger in-group bias in the competitive as compared with a neutral priming condition and a no priming control condition as indicated by a greater difference between the donated stickers with in-group and out-group members. In the neutral priming and a non-priming control condition, children did not show any in-group bias, and their donations were mostly equal between the groups. Importantly, the stories used for priming were not related to the established groups suggesting a spill-over effect of the primed competition to the intergroup scenario. Thus, a competitive as compared with a neutral orientation (e.g. induced through priming) increased children's in-group bias.

However, in Spielman's neutral priming story, two children play together on a playground, giving it a somewhat cooperative and not entirely neutral touch. According to this view on the priming stimuli, the results suggest that competitive priming increases in-group bias, while cooperative priming and the non-priming condition do not show any in-group bias. This interpretation would imply that in intergroup contexts primarily competition increases in-group bias, while the promotion of cooperation does not have particularly beneficial effects on reducing in-group bias. This conclusion speaks against research stressing the importance of cooperation to reduce in-group bias [32–35]. A systematic investigation of the effects of a competitive, cooperative and neutral orientation can help us to better understand the distinct effects of these orientations on children's in-group bias.

## 1.4. The effects of cooperation and competition on prosociality

Besides affecting in-group bias, cooperative and competitive orientations are assumed to affect children's general willingness to act prosocially regardless of a recipient's group membership. A cooperative orientation comprises anticipated prosociality of others and an increased prosociality toward these [3]. A competitive orientation is characterized by anticipated resistance and decreased prosociality [4]. Neutral contexts, in which agents' goals are independent, do not change the expectation of others' prosociality [5].

Past research corroborates these predictions. Preschoolers' prosocial behaviour toward interaction partners is more likely within cooperative than competitive contexts [36–43]. Cooperative compared with competitive [36,40,44–48] and neutral contexts [24,49,50] increased preschoolers' prosocial behaviour toward their previous interaction partners even in subsequent interactions. Further, cooperation and competition affect preschoolers' sharing of resources unrelated to previous cooperation or competition. The involvement in an alleged drawing contest as opposed to a non-competitive context decreased 4- to 6-year-olds' sharing of both related (i.e. crayons) and unrelated resources (i.e. stickers; [51]). Similarly, cooperation promoted 3- to 5-year-olds' sharing of an unrelated resource compared with a control context with no interdependence (i.e. sharing more candy after cooperatively retrieving a toy; [52]). However, Plötner et al. [24] find preschoolers' sharing of unrelated resources to be unaffected by previous cooperative interactions.

Further, children share more resources with uninvolved third-parties after experiencing long-time cooperative gaming interventions as compared with control conditions comprising regular education practices [53–55]. Toppe et al. [56] examined the short-term effect of different interdependent contexts on prosociality directed toward a third-party. In the study, dyads of 4- to 5-year-old children played a game either cooperatively, competitively or solitarily to elicit the respective psychological orientation. Children's sharing with and social inclusion of a third-party, as well as the free play of co-players, were assessed after 5 min of play. Children shared more resources after playing a cooperative as compared with a competitive game. Children's social inclusion and prosociality in free play were not affected by the different contexts of the game.

While these findings point to the importance of a cooperative psychological orientation on children's sharing, methodological limitations still hinder conclusions. First, although it was included as a statistical control variable, the study design by Toppe et al. [56] did not experimentally control the outcome of the game (i.e. winning or losing). However, controlling the game's outcome might allow more robust conclusions on the effect of different gaming contexts since the dynamic of the game would be kept constant between subjects. Second, Toppe et al. [56] did not consider children's engagement in the

game as a potential predictor for their subsequent prosociality. It seems plausible that children who are more engaged while playing a game absorb the respective social context more strongly. For example, players who strongly engage in a cooperative game cooperate more while playing this game than low engaging players. Consequently, high engaging players might have a more pronounced cooperative orientation than low engaging players and, thus, act more prosocially afterward. One can predict a respective effect for competitive games. Hence, children's engagement might interact with the different forms of interdependence and account for inter-individual differences within experimental conditions, and should be considered as a predictor for children's prosociality. Finally, Toppe *et al*. [56] used a highly interactive game to elicit a cooperative and competitive psychological orientation. That is, children needed to constantly coordinate their actions with their co-players in order to be successful. This high demand for coordination between co-players is similar to most of the previous studies, investigating the effect of cooperative and competitive games [24,36,37,41,44,46,49,50,52,54,55], but see [51]. However, social interdependence theory states that the relation of goals is the main driver of the predicted effects regardless of whether players need to actively coordinate their actions [1,2], with cooperation to promote, and competition to lower prosociality. Reducing the demand for coordination might help to learn more about the mere influence of the relation of goals on children's prosocial behaviour.

## 1.5. The current study

In sum, past research suggests that cooperation and competition can elicit psychological orientations influencing children's in-group bias and prosocial behaviour. The elicitation of a cooperative orientation (e.g. through priming or playing a game) might be a promising intervention on both children's intergroup and prosocial behaviour. On the one hand, a cooperative orientation might reduce preschoolers' in-group bias. On the other hand, it might promote preschoolers' general prosociality toward others. The current study aimed to examine these two effects and to replicate the findings of Spielman [31] and Toppe *et al*. [56]. Like Toppe *et al*. [56], we used a dyadic game (after this referred to as an intervention game) to elicit a cooperative, competitive and solitary orientation. After playing the intervention game, we assessed 4- to 6-year-old children's sharing and social inclusion behaviour in a minimal group situation.

To measure their sharing, children could divide stickers between themselves and recipients having different group memberships (in-group versus out-group member). We coded the number of shared stickers for each recipient. The social inclusion of out-group members into an in-group interaction was measured with a ball-tossing task similar to the one used by Toppe *et al*. [56]. In the current study, we used a modified version of this paradigm (see [22]), in which children played a ball-tossing game with an in-group puppet. Throughout this tossing game, an out-group puppet approached the two in-group members asking to join the game. We coded whether and to what extent children included the approaching out-group puppet and how they want their in-group member (i.e. the in-group puppet) to behave in this task.

Given the urgent need for replications in psychological research [57], our study aimed at a conceptual replication of the results found by Spielman [31] and Toppe *et al*. [56] with a larger sample size. Besides, we intended to extend these two studies in different aspects.

Spielman [31] did not systematically distinguish the effect of a cooperative orientation on children's prosociality. So far, we do not know whether the effects of a cooperative and a solitary context on preschoolers' in-group bias are similar or not. A systematic investigation of all three contexts— cooperative, competitive and solitary—is needed to evaluate Spielman's effects. Notably, we slightly changed the order of Spielman's procedure in the current study. In Spielman's study, children have been assigned to a group before the orientation has been elicited. Children were first assigned to a minimal group, then primed with the story, and finally distributed stickers. In the current study, group membership was established *after* playing the intervention game to minimize the chance that dyad partners would know their partner's group membership. Also, we wanted the game to be independent of the groups to learn more about the impact of cooperative and competitive orientations on novel subsequent intergroup contexts.

In contrast with Toppe *et al*. [56], we experimentally controlled the outcome of the intervention game and considered children's engagement while playing the game as a predictor for their subsequent prosociality. Stronger engagement in the game might result in a stronger effect of the respective context. We measured children's engagement in the intervention game through their physical effort while playing. Besides this potential moderating effect, we tested whether children's engagement in

the intervention game would be the highest in the competitive context since social comparisons are assumed to be more salient in competitive than in cooperative and solitary contexts [58,59].

Further, Toppe et al. used a highly coordinative game which might have confounded the mere effect of goal relations with the effect of coordinating with interaction partners. Thus, we used a non-interactive game to elicit the respective psychological orientations to learn more about the mere effect of goal relations on children's prosocial behaviour.

Finally, we aimed to learn more about the impact of children's socialization context on their in-group bias. To contribute to this line of research, we surveyed children's parents and assessed their social dominance orientation (SDO) [60]. SDO is a promising proxy for parents' intergroup socialization of their children, being one of the strongest predictors of parents' own intergroup attitudes and behaviour [61]. SDO is assumed to be a stable trait indicating the preference for inequality among social groups [60,61]. People scoring high on SDO tend to perceive the world as competitive, leading them to feelings of dominance and superiority [62]. In adults, SDO is strongly linked to diverse intergroup behaviours [60,61,63]. Among others, SDO is related to prejudice toward familiar groups and affiliation toward minimal groups, with adults scoring higher on SDO showing a stronger in-group preference (e.g. [60,64,65]). Furthermore, adults who score high on SDO tend to approve of unequal resource distributions [66].

For the link between parental intergroup attitudes and children's social behaviour, mixed findings exist, with studies indicating positive [67–70], negative [68] and no connections [68,71,72]. In their meta-analysis, Degner & Dalege [73] found a moderate positive correlation ($r = 0.38$) between parents and their children's intergroup attitudes. Previous studies also found significant associations between adolescent children's and their parents' SDO [74,75], whereas only a weak association was found for 8- to 10-year-old children [76]. Further, 4- to 5-year-olds were more likely to punish an in-group member who acted unfairly in an intergroup context when their parents' SDO was lower, indicating an increased sense of fairness in children whose parents score low on SDO [77]. These findings suggest that parental SDO appears to be related to their preschool-aged children's intergroup behaviour. Thus, in addition to Spielman [31] and Toppe et al. [56], we assessed parents' SDO to predict their children's sharing and social inclusion behaviour in an intergroup context to learn more about the impact of children's socialization contexts on these behaviours.

Past research led us to the following hypotheses (for preregistration, see osf.io/ay8hm):

First, children would show an in-group bias such that they share more with an in-group than with an out-group member across all experimental conditions. We further investigated how the different gaming contexts (cooperative, competitive and solitary) would shape the differences between the stickers shared with the in- and the out-group member and their social inclusion behaviour toward an out-group member. Second, children's total number of shared stickers would be influenced by the different gaming contexts, with more shared stickers after playing a cooperative game as compared with a competitive or a solitary game. Playing a competitive game compared with a cooperative or a solitary game would lead to fewer shared stickers. Third, children's engagement while playing the intervention game would be higher in the competitive context as compared with the cooperative and solitary context. We further explored how children's engagement would moderate their in-group bias and general prosociality. Fourth, parents' SDO would be positively related with in-group bias for children's sharing and negatively related to the general willingness and speed of children's inclusion of an out-group member.

For all hypotheses, we investigated how far children's age accentuates the effect of the experimental conditions by considering the interaction between condition and age. The consideration of this interaction term seems highly relevant as the understanding of cooperative and competitive contexts seems to emerge throughout preschool age [78–82].

Initially, we planned to test the impact of the contexts of the intervention game, age, engagement and parents' SDO on each dependent variable in one single model. However, the parental survey response rate was relatively low (42%; $n = 54$; see Materials and methods). Therefore, we decided to deviate from our preregistration and investigated our fourth hypothesis in a separate data analysis with the respective subsample (see Data analyses). We decided for this deviation to ensure statistical power for the effects of the other predictors.

In addition to our preregistration, we also explored whether the duration of playing the game moderated the potential effect of the different gaming contexts. It might be that the expected effects of the gaming contexts become more pronounced after playing the game for a while as children's reactions to the contexts might be delayed. The idea for this explorative analysis occurred in the review process and was integrated into our preregistered analysis.

# 2. Material and methods

## 2.1. Participants

The sample used for analysis consisted of 144 German children aged between 4 and 6 years (50% female; mean age = 4.96 years; age range = 4.03 to 6.05 years). Children were from a mid-sized German city, and recruitment was based on a laboratory-maintained database, including children from about 150 day-care centres. Participants tested in this study were from 20 day-care centres located in different districts of the city, allowing the assumption that children had diverse socio-economic backgrounds. We aimed to include age as a continuous variable into the statistical analysis and, thus, wanted children's age to be distributed evenly. To achieve a relative even distribution of age across conditions, about one half of the sample should be aged between 4 and 5 and the other half aged between 5 and 6 within each condition ($n_{4y/o} = 78$, $n_{5y/o} = 63$, $n_{6y/o} = 3$; for a histogram of children's age, see electronic supplementary material, figure S1).

Children participated in same-sex dyads. In each dyad, children were aged equally such that the maximum difference between children's age was not larger than six months. Besides the criteria of age and sex, dyads were determined randomly. As children were tested in their day-care centre, they were often familiar with their co-player but dyad constellation was not influenced by children's relationship with each other.

The sample size of $N = 144$ was suggested by a prior power analysis expecting a medium effect with a statistical power of 0.80 and a probability of 0.05 for a type I error. This power analysis referred to the overall effect of the variables of interest—interaction between condition and age and the main effects of the condition, age and engagement. The power analysis was conducted with the *pwr* package [83] in R statistical software ([84]; see R script at osf.io/pu89t/).

In the current study, we investigated the effects of cooperation and competition on German children's prosocial behaviour in an intergroup context. In Germany and other Western societies, children typically grow up with specific experiences related to groups, social interdependence and fairness. For example, US adults show stronger in-group biases and assume more intergroup competition in minimal group contexts as compared with adults from (non-Western) Japan [85]. Parents in Western societies typically scaffold and reward cooperative interactions with their children from early in ontogeny [86,87] and, at the same time, believe that their children need to learn how to get on within competition [88]. Further, in Western societies, the consideration of merit crucially influences children's sharing behaviour and sense of fairness. German children were found to distribute spoils based on their own and others' merit in earning these, whereas children from African gerontocratic or hunter–gatherer societies applied different sharing heuristics [89]. Likewise, children from Western societies shared more with hard-working as compared with less-working peers [90,91]. Hence, the partner's engagement during cooperative endeavours seems particularly relevant for German children's subsequent sharing decisions. Finally, in Western cultural contexts, parents are assumed to be an essential part of children's intergroup socialization [76,92] and one can expect a substantial influence of parents' SDO on children's intergroup behaviour.

The study was part of a project that has been approved by the ethics committee of the Medical Faculty of Leipzig University (project name: 'Non-pathological development of social behaviours and competences in children and adults with behaviour-based observational, peripheral physiological and psychometrical methods'; protocol number: 169/17-ek). For all children, parents gave informed consent for participation. Testing took place in the participants' day-care centres. An additional five dyads were tested but excluded from data analysis due to reluctance to participate (four dyads) or experimenter error (one dyad). Due to restricted visibility in video recordings, gaming engagement could not be coded for one dyad in the competitive condition. The sharing behaviour of three children was excluded due to a failed comprehension check. Due to technical problems, the sharing behaviour of four children was not recorded, but live coded behaviour was used for data analyses.

## 2.2. Materials

Children played an intervention game in which they could rotate a tube to allegedly manipulate the course of a train (figure 1a). Tubes were made of plastic material, fixed in a wooden mount and had rubber naps for grip. Tube apparatuses were taped on the ground. The automotive train operated on wooden tracks, and the course of the train included a switch roofed by a cover made of cardboard. The switch could be operated with a stick protruding out the cover, but still being hidden from

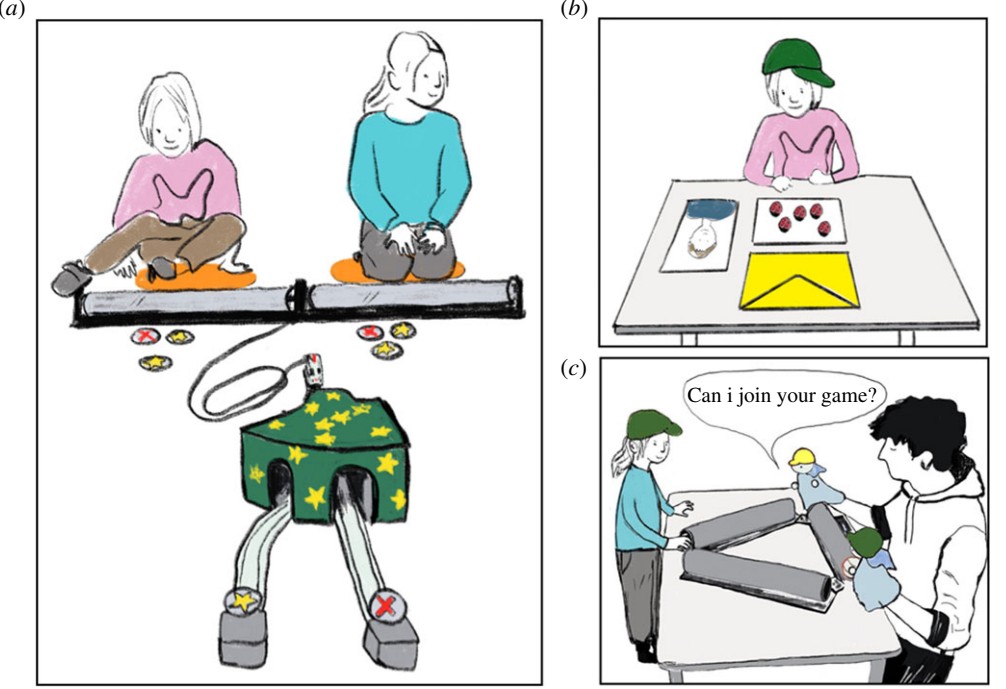

**Figure 1.** In (*a*), the apparatus for the intervention game is depicted (arrangement for the cooperative condition). Panels on the right show (*b*) the set-up for the Dictator Game and (*c*) the apparatus used in the social inclusion task.

participants' sight. The switch split the track of the train toward two ends. Depending on the condition, one or two cables connected the tubes with the switch, and laminated stars or crosses were placed on plastic holders at the end of the tracks. Stimuli for the dictator games were four portraits (two depicting a girl and two a boy) with a happy facial expression taken from the NIMH Child Emotional Faces Picture Set [93]. In each dictator game, children received five identical stickers, which they could put into coloured envelopes (figure 1*b*). In the social inclusion task, we used a triangle of plastic tubes fixed on a wooden frame and a rubber ball that could run through the tubes (figure 1*c*). Also, four hand puppets (two depicting a girl and two a boy) with green and yellow caps and scarves were used. To establish group membership, children were equipped with green caps.

## 2.3. Design and procedure

After consultation with a child-care worker, the experimenter asked children with parental consent whether they wanted to play a game with another child. When two children of a similar age agreed, the experimenter guided the two children to the test room which was a freely available room in the day-care centre.

We randomly assigned dyads to one of three experimental conditions: cooperative ($n = 48$ children), competitive ($n = 48$ children) and solitary ($n = 96$ children). Importantly, we tested twice as many children in the solitary condition ($n = 96$ children) since only one child interacted with the gaming apparatus (targets; $n = 48$ children) while the other child was parallelly engaging in a non-gaming activity (observers; i.e. drawing a picture; $n = 48$ children; details see below).

In deviation from our preregistration, we only analysed the data of those children in the solitary condition who played the intervention game (targets; $n = 48$ children) and excluded the data of children who did not play the intervention game (observers; $n = 48$ children) from our inferential analyses. Our initial idea was to analyse the data of the observers as well since these might constitute a non-gaming baseline. However, most observers oriented toward the intervention game frequently and were slightly frustrated that their partner (but not they) could play the game. From our view, this situation does not constitute a baseline. Besides, we did not have any prediction on how this particular social comparison affects children's subsequent prosociality. Thus, we decided to exclude these children from the data analysis but report the descriptive results of this subsample. Importantly, already in the conception of the study, the observer sample was planned as an additional sample.

That is, our prior power analysis (which determined the sample size of the three central conditions) was not planned with the observer condition.

Dyads played the intervention game in two phases with the same context (between-dyad design). Dependent measures were children's engagement while playing the game, sharing with in- and out-group members, and social inclusion of an out-group member into an in-group interaction. Two experimenters conducted the study. A detailed description of the procedure including the exact wording is available at osf.io/pu89t/.

## 2.4. First gaming phase

In each condition, two experimenters introduced the intervention game with a different context. In the cooperative condition, children's tubes were placed directly beside each other, and a single cable ran from the tubes to the covered switch. At one end of tracks a star and on the other a cross were placed. The first experimenter (E1) told the children that the game is played cooperatively and that both would win or lose together. Children were told that they needed to win more stars than crosses in order to win the game and that they would lose if they received more crosses than stars. In each round, the train started to drive from the starting position, and the final position (either the end with star or cross) determined the outcome of the respective round. Children were told that they could influence the course of the train using the tubes in front of them: if both rotated the tubes fast enough, the train would drive to the star. If not, the train would end at the cross. Before children started to play, E1 and the second experimenter (E2) demonstrated two rounds of the game. In the first demonstration round, both experimenters turned their tubes slowly, and the train drove to the cross. In the second demonstration round, the experimenters turned their tubes fast, and the train ended at the star (E1 secretly changed the switch when placing back the train to the starting position). Hereafter, dyads played the game for eight rounds. After each round, both experimenters stated the outcome (The train drove to the star/cross.), placed the respective token in front of the tubes and placed a new token at the respective end of the track. The game always ended in an equal number of stars and crosses (four of each), and the order of winning and losing throughout these eight rounds (hereafter referred to as course of the game) was experimentally controlled.

In the competitive condition, the two tubes were placed slightly oblique, and from each tube, a cable ran to the switch. On both ends of the track, a star was placed. Children were instructed that the game would be played against each other and that if one of them won the game, the other one would lose. The player who collected more stars would win the game. Then, E1 explained that in every round, the train would start to drive from the starting position and that the final position would determine the outcome of the respective round: the participant on whose side the train ended would receive a star. Children were told that they could influence the course of the train by rotating their tube: when turning the tube faster than their opponent, the train would drive to their side. E1 and E2 explained this with two demonstration rounds. In the first demonstration round, E1 turned her tube while E2 did not, and the train drove to E1's side. In the second demonstration round, E2 turned her tube while E1 did not, and the train drove to E2's side (E1 secretly changed the switch when placing back the train to the starting position). Hereafter, the children played the game for eight rounds. After each round, both experimenters stated the outcome (The train drove to the side of *Name of Child*.) and placed the star in front of the participant. The game always ended in an equal number of stars for each player (four each). There were two courses of the game (i.e. the order of winning and losing over the eight rounds) that were experimentally controlled.

In the solitary condition, one child played the game (target child), while the other child (observer) drew pictures parallelly. Here, only one tube was used. At the end of the tracks, a star and a cross were placed. E1 introduced the game to the target child. Rules were exactly the same as in the cooperative condition, with only one modification: the target child received stars and crosses solitarily, while the dyad partner was not involved in the game. There were two courses of the game that were experimentally controlled and always ended in an equal number of stars and crosses (four of each). While the target child played the train game, E2 equipped the observer child with crayons and papers. E2 explained that they could draw a picture while E1 and the target child would play something different. Then, E2 pretended to work on something and gave suggestions if children had no ideas for their drawings. If children observed the target child playing the train game, E2 guided their attention back to their drawing.

After the first gaming phase (duration approximately 5 min), one of the experimenters left the room with one participant (P1) and went to a quiet place in the day-care centre (e.g. other room or empty

corridor), while the other experimenter and the second participant (P2) stayed in the test room. The roles of P1 and P2 were assigned randomly.

## 2.5. Group assignment

After the first gaming phase, the experimenters assigned both participants to a minimal group in separate rooms. Importantly, participants did not know that their co-player was assigned to a group. Experimenters mentioned that there were two groups (green and yellow), looked into a bag, uncased a green cap and stated that the participant would be a member of the green group. Children were always assigned to the green group, but group allocation appeared to be random. Children received a cap, and the experimenters stressed their group membership. Similar procedures have been used in previous studies to manipulate children's group membership (e.g. [17,22,94]).

## 2.6. Dictator game

P1 participated in two consecutive dictator games. The experimenter introduced two same-sex peers while placing two portraits in front of P1. The experimenter explained that one child belonged to the green (in-group member) and the other to the yellow group (out-group member), which was indicated by respectively coloured envelopes placed in front of the portraits.

To ensure comprehension, children needed to name their own group colour and whether they share group membership with the portrayed peers. Notably, three children could not identify colours and groups correctly and were excluded from the analysis.

Then, the experimenter moved either the portrait and envelope of the in-group or the out-group member away so that children only saw one depicted peer (counterbalanced). Children were given five identical stickers and an instruction to share these with the peer by putting the stickers into the coloured envelope. The stickers participants wanted to keep for themselves could be placed in a second envelope close to the child (coloured brown). While distributing the stickers, the experimenter turned around and did not observe the child. To ensure that participants understood the instruction, they were asked four questions before dividing the stickers. They were asked to whom the stickers belonged; where they could place stickers they want to share; where they could place stickers they want to keep for themselves; whether anyone could see them while placing the stickers. The experimenter repeated the respective information one more time if the children answered a question incorrectly. Three children did not pass the comprehension check in the dictator game, and their sharing behaviour was excluded from statistical analysis. The second dictator game with the other portrayed peer followed the same procedure so that all children shared with an in-group and an out-group member.

## 2.7. Social inclusion

Parallel to the dictator game, the other experimenter conducted a social inclusion task with P2 in the test room. After the group assignment, the experimenter operated two hand puppets and introduced these to P2. Both puppets matched the participant's sex. The in-group puppet (wearing a green cap and scarf) was introduced first, followed by the out-group puppet (wearing a yellow cap and scarf). Puppets asked for the child's name, told their names and the group they were assigned to, and stressed that they were either in the same (in-group puppet) or in a different group (out-group puppet). Then, the experimenter placed both puppets in front of P2 and repeated the child's and the puppets' group membership. The experimenter moved the in-group puppet close to the child while stating that they were members of the same group (Both of you are in the green group.). The out-group puppet was placed further away, and the experimenter stressed that this puppet belonged to a different group (She/He is in the yellow group.).

To ensure comprehension, children had to name their group, the group of the puppets, and state whether they share group membership with the puppets. Respective information was repeated one more time by the experimenter if children failed to answer one of these three questions. All children passed this comprehension check.

Hereafter, the in-group puppet introduced a ball-tossing game and revealed the covered apparatus. The in-group puppet and the child passed the ball back and forth through each of the three tubes of the apparatus. The in-group puppet stayed at one corner of the apparatus (counterbalanced) and initiated another two rallies. When the in-group puppet held the ball, the out-group puppet appeared at the vacant corner of the triangle stating 'Hello'. While holding the ball, the in-group puppet

decided to pass the ball to the child after speaking with itself and thinking aloud about to whom it would pass the ball to (Do I pass the ball to *Name of out-group puppet* or to *Name of child*?). Children could freely decide to which of the puppets they pass the ball. Both puppets always passed the ball to the child. If not included for two consecutive rallies, the out-group puppet gave a prompt indicating the desire to be included when the in-group puppet held the ball (Can I join your game?). Again, the in-group puppet decided to pass the ball to the child after weighing both alternatives and thinking aloud.

Four rallies were played in this way, followed by a final directive trial, in which the in-group puppet addressed the child directly and asked to whom it should pass the ball (to the out-group puppet versus the participant; 'To whom should I pass the ball? To you or to *Name of out-group puppet*?').

When children were holding the ball, the experimenter avoided eye-contact and faced the floor. If children did not pass the ball for about 10 s, the in-group puppet encouraged them to pass the ball (Now, it is your turn.). A demonstration video of this task is available at osf.io/pu89t/.

## 2.8. Second gaming phase

After the first assessment of children's sharing and social inclusion, children gave their caps back to the experimenters and reconvened in the test room. Again, children played the intervention game with the same rules as in the first gaming phase after E1 shortly brushed up the rules. In the cooperative and competitive condition, children changed tubes. In the solitary condition, children's roles remained the same, and the target child played the intervention game a second time. Children played the game in a second phase to minimize the interference of the assessment of the prosocial behaviours and to strengthen the impact of the different contexts of the intervention game. Also, without a second gaming phase, the effect of the intervention game might be weaker for the latter assessed prosocial behaviour (for a similar procedure, see Toppe *et al.* [56]).

Similar to the first game phase, the game ended after eight rounds. Then, children participated in the social inclusion task and dictator games. Now, children changed roles so that for all children, both social inclusion and sharing were assessed. Before the second assessment of the dependent variables, experimenters refreshed the group assignment and asked the participants about their group membership to ensure comprehension. Then, sharing and social inclusion were assessed with the same procedure described above.

Dyads in the solitary condition played the intervention game cooperatively after the assessment of all dependent variables so that children who only observed their peers in the testing phase could also play the game.

## 2.9. Online survey

After their participation, children received an envelope containing the stickers they wanted to keep for themselves. Further, the envelope contained an information letter about the procedure, the study's goals, and a printed version of the social dominance scale by Ho *et al.* [66] (table 1). Items of the social dominance scale were statements related to groups. Participants can indicate their agreement with these statements on a 7-point Likert-scale (ranging from *strongly oppose* to *strongly favour*). In the letter, parents were asked to fill out the survey and to send it to the Department of Early Child Development and Culture at the Faculty of Education, Leipzig University. Postal charges were prepaid and printed on enclosed envelopes for parents' reply.

We translated the original English scale into German and double-checked our translation with an independent native speaker of both languages. In the translation process, only minor disagreements occurred which were resolved by discussing the respective passages. Besides the social dominance scale, the survey comprised other scales as a pilot for upcoming projects.

The information letter explicitly stated that participation was voluntary and that all data were analysed anonymously. In addition, the letter showed a QR-code with a link to an online version of the survey. Parents' reports could be assigned to children's data by an individual ID code on the letter. In total, $n = 54$ parents participated in the survey. All data are publicly available at osf.io/pu89t/.

## 2.10. Coding and reliability

Sessions were videotaped with two camcorders. Coding was done live and from video by the first author. To measure gaming engagement for each gaming phase, we coded the number of tube rotations in each round. For some children ($n = 17$), not all rounds could be coded due to limited

**Table 1.** Social dominance scale.

| item | M (s.d.) |
|---|---|
| an ideal society requires some groups to be on top and others to be on the bottom[a] | 3.48 (1.69) |
| some groups of people are simply inferior to other groups | 3.61 (1.90) |
| no one group should dominate in society[a] | 1.48 (0.72) |
| groups at the bottom are just as deserving as groups at the top | 2.44 (1.46) |
| group equality should not be our primary goal | 3.89 (1.73) |
| it is unjust to try to make groups equal | 3.29 (1.89) |
| we should do what we can to equalize conditions for different groups | 1.96 (0.99) |
| we should work to give all groups an equal chance to succeed | 1.76 (1.06) |
| merged scale | 2.82 (1.06) |

[a]Items were excluded and not part of the merged scale. Note. Scale reached from 1 to 7 with high values indicating a strong SDO.

visibility (range of missing values = 1 to 8). For the dictator game, we coded the number of shared stickers with the in-group and the out-group member. For the social inclusion task, we coded whether participants included the approaching out-group puppet within the four rallies at least once; in which rally participants included the out-group puppet the first time; the total number of passes to the out-group member; and the chosen option in the directive trial. Two coders blind to hypotheses coded a random quarter of the data. Inter-rater reliability was excellent for children's sharing and social inclusion behaviours (all Cohen's $\kappa = 1$) and excellent for the number of rotations while playing the intervention game (all $ICCs > 0.95$, all $ps < 0.001$).

We aimed to aggregate all items of the social dominance scale to attain a single index of parents' SDO. To establish unidimensionality, we performed a confirmatory factor analysis and inspected modification indices in case of model misfit indicated by a comparative fit index (CFI) < 0.95, root mean square error of approximation (RMSEA) > 0.08, and a standardized root mean square residual (SRMR) > 0.08. If modification indices suggested correlated residuals, we checked the involved items for redundancy and removed one of the redundant items; specifically, the one with the lower factor loading. Further, we inspected the loading pattern for signs of missing relatedness to the construct of single items. If single loadings had less than half the size of the average loading, the respective item was removed.

Following this preregistered procedure, we excluded two items of the social dominance scale to reach the preregistered criteria (table 1). This final scale had a satisfactory model fit (CFI = 1.000; RMSEA > 0.001; SRMR = 0.046) and good reliability (McDonald's $\omega = 0.820$).

## 2.11. Data analyses

To statistically test our hypotheses, we fitted generalized linear mixed models (GLMMs; [95]) in R statistical software [84] using the *lme4* package [96]. For all models, we compared a full model comprising all predictors of interest and control variables with a null model comprising control variables only. Only in the case of a significant full-null model comparison, we tested the effect of individual predictors by comparing the full model with reduced models lacking them one at a time. By such full-null model comparisons multiple testing can be avoided [97]. in the case of non-significant interactions, we continued our analysis with a model comprising the related main effects only. For all model comparisons, we conducted likelihood ratio tests.

Further, we calculated Bayes factors (BFs) comparing the evidence for the full model with the evidence for the null model given our data. BFs are a numerical value quantifying how well a hypothesis predicts the given data relative to a competing hypothesis [98]. Here, we used the *bayestest* R package [99] to extract the BF for the full and the null model and applied Jeffrey's system [100] to interpret these. The advantage of BFs is that they allow for a differentiated interpretation of the support of the null versus the alternative hypothesis.

All GLMMs included age (measured in days) as a continuous variable and dyad identification number as a random intercept effect to control for within-dyad variance. The GLMMs for the number of rotations, children's sharing, first inclusion, number of passes and directives revealed singular fits,

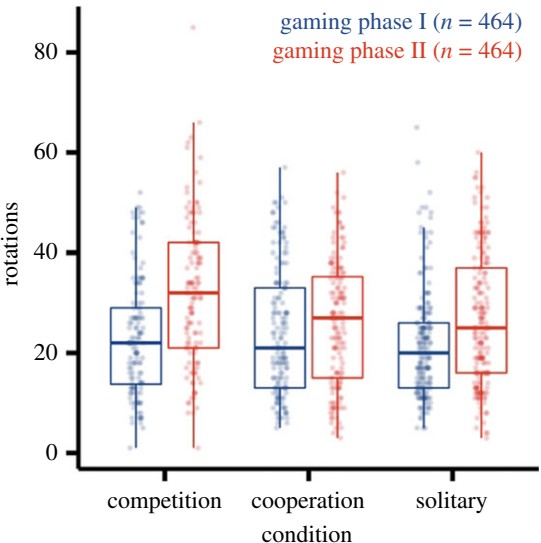

**Figure 2.** The number of rotations in the intervention game across conditions and gaming phases.

meaning that some cells of the estimated variance–covariance matrix have been estimated as exactly zero [96]. This can occur in multi-level models, and in such a case, likelihood ratio tests may be inappropriate to determine the significance of a predictor [96,101]. When a GLMM computed with the lme4 package suggested a singular fit, we used a maximum penalized likelihood approach, which is a partially Bayesian method using regularizing priors (*blme* package; [101]). When using the *b(g)lmer* function of this package, model comparisons are valid to detect the significance of predictors. Notably, the results between the models calculated with the *lme4* and the *blme* package differ marginally and did not change the interpretation of the significance of any predictor. The script for analyses is publicly available at osf.io/pu89t/.

Please note that this data analytic approach slightly deviates from our preregistered analysis. In response to a reviewer's comment, we tested the effect of the interaction between condition and the duration of the game (i.e. gaming phase). This interaction was not part of our preregistered analysis. However, we acknowledged the reviewers' advice in order to gain a better understanding of the effect of the gaming contexts on children's social behaviour. Due to the extended number of tests created by the additional interaction term, we decided to counter multiple testing by full-null model comparisons. Also, our preregistered analysis did not include BFs.

## 3. Results

### 3.1. Engagement

In the GLMM analysing children's gaming engagement, we added the interaction of age and condition and the interaction between condition and gaming phase as predictors while controlling for the main effects of participants' sex and trial number. Further, we included subject as a random intercept to account for within-subject differences. The model was fitted using a Gaussian error distribution.

Overall, the full-null model comparison was clearly significant, $\chi^2_8 = 60.932$, $p < 0.0001$. Pairwise model comparisons revealed a significant effect of the interaction between condition and gaming phase, $\chi^2_2 = 32.081$, $p < 0.0001$, while interaction between age and condition did not reach significance, $\chi^2_2 = 1.150$, $p = 0.563$. Thus, we conducted pairwise comparisons with a model comprising the main effect of age, the interaction between condition and gaming phase, and controls. Again, the interaction between condition and gaming phase reached significance, $\chi^2_2 = 32.081$, $p < 0.0001$. Inspection of the descriptive results suggest that the number of tube rotations increased from the first to the second gaming phase across all conditions but that this increase differed between conditions (figure 2). More precise, children showed a stronger increase in their tube rotations in the competitive condition ($\Delta_{\text{Competition}} = 9.273$) as compared with the other two conditions ($\Delta_{\text{Cooperation}} = 1.882$; $\Delta_{\text{Solitary}} = 4.924$). Also, we found a significant effect of age on the number of rotations, $\chi^2_1 = 5.430$, $p = 0.020$, such that children showed more tube rotations with increasing age. Sex and trial did not seem to affect the

**Table 2.** Results for children's sharing.

| | recipients' group membership | |
| --- | --- | --- |
| | in-group | out-group |
| condition | *M* (s.d.) | *M* (s.d.) |
| cooperative | 1.30 (1.26) | 1.37 (1.22) |
| competitive | 1.26 (1.24) | 1.30 (1.21) |
| solitary | 1.42 (1.41) | 1.16 (1.22) |
| observer[a] | 1.20 (1.28) | 1.20 (1.33) |

[a]Children who observed the target child in the solitary condition were excluded from all inferential analyses.

number of rotations, $p$s > 0.668 (details in electronic supplementary material, table S1). The BF for the model comparison between the full model and the null model suggested decisive strength of evidence in favour of the full model (BF > 1000).

## 3.2. Sharing

To examine whether condition and age affect children's sharing behaviour, we ran a GLMM with children's sharing with in-group and out-group members as dependent variables. As fixed effects, we included the three-way interaction between condition, age and group membership of the recipient (in-group versus out-group), the two-way interactions between condition and gaming engagement and condition and gaming phase. Participants' sex, gaming course (order of wins and losses), as well as order of sharing (in-group versus out-group first), were included as control variables. The model was fitted using a Poisson error distribution.

Overall, a full-null model comparison did not reveal a substantial effect of the variables of interest, $\chi^2_{17} = 15.471$, $p = 0.562$ (table 2; details in electronic supplementary material, table S2). The BF related to this model comparison suggested strong evidence in favour of the null model (BF < 0.001).

## 3.3. Social inclusion

To test whether children's social inclusion of out-group members differed as a function of age, condition and children's engagement, we conducted four GLMMs for the behaviours coded in the social inclusion task. All these models included the two-way interactions between condition and age, condition and gaming engagement, as well as the two-way interaction between condition and gaming phase. We controlled for participants' sex, gaming course and the position of puppets (in-group puppet left versus right). The models for the rally of first inclusion and the number of passes to the out-group puppet were fit using a Poisson error distribution. Models analysing whether participants included the out-group puppet at least once and their decision in the directive trial were fit using a binomial error distribution.

For none of these social inclusion behaviours, the full-null model comparison revealed an overall effect of the variables of interest, $\chi^2_{11} = 14.261$, $p = 0.219$ for general inclusion; $\chi^2_{11} = 4.541$, $p = 0.951$ for first inclusion; $\chi^2_{11} = 12.561$, $p = 0.323$ for number of passes; $\chi^2_{11} = 10.345$, $p = 0.500$ for directives (table 3; details in electronic supplementary material, table S3). All BFs related to these full-null model comparisons suggested strong evidence for the null models (all BFs < 0.0001).

## 3.4. Social dominance orientation

The descriptive statistics of the social dominance scale are given in table 1. For the subsample of children whose parents filled out the social dominance scale ($n = 54$), we examined how the interaction between parents' social dominance orientation and the condition is associated with children's sharing and social inclusion behaviour. All models were exactly the same as compared with the previous analysis but further comprised the respective interaction term. To indicate the significance of this interaction, we compared the full model with a model lacking the interaction term and calculated BFs comparing the evidence for the full model with the evidence for a null model given our data.

**Table 3.** Results for children's social inclusion.

| condition | general inclusion % including out-group | first Inclusion M (s.d.) | number of passes M (s.d.) | directive % including out-group |
|---|---|---|---|---|
| cooperative | 54.17 | 1.77 (1.14) | 1.04 (1.13) | 56.25 |
| competitive | 63.04 | 1.45 (0.78) | 1.33 (1.14) | 56.52 |
| solitary | 62.50 | 1.47 (0.82) | 1.15 (0.97) | 64.58 |
| observer[a] | 56.25 | 1.70 (1.03) | 1.00 (0.99) | 68.75 |

[a]Children who observed the target child in the solitary condition were excluded from all inferential analyses.
Note: General inclusion refers to whether participants included the out-group puppet at least once in the four trials. First inclusion refers to the rally in which participants included the out-group puppet the first time (coded with 1 to 4). Number of passes refers to passes to the out-group puppet (coded with 0 to 4). Directive refers to whether participants stated that the first puppet should pass the ball to the out-group puppet or themselves.

These analyses did not indicate any evidence that parents' social dominance orientation moderates the effect of the experimental conditions on children's prosocial behaviours; $\chi_2^2 = 1.782$, $p = 0.410$ for sharing; $\chi_2^2 = 0.225$, $p = 0.893$ for general inclusion; $\chi_2^2 = 0.713$, $p = 0.700$ for first inclusion; $\chi_2^2 = 1.040$, $p = 0.595$ for number of passes; $\chi_2^2 = 0.475$, $p = 0.789$ for directives. All BFs related to full-null model comparisons suggested strong evidence for the null hypothesis (all BFs < 0.0001).

# 4. Discussion

The current study investigated how a cooperative, competitive and solitary game context affects German preschoolers' intergroup behaviour in a minimal group context and their prosociality toward others more generally. After playing a game in either a cooperative, competitive or solitary fashion, we assessed children's sharing with an in-group and an out-group member as well as their social inclusion of an out-group member into an in-group interaction. The three contexts of the game did neither influence children's intergroup behaviour nor the general level of prosociality when sharing with a third-party. For the physical engagement while playing the game, we found that children's engagement increased over the course of the game and that this increase was most accentuated when playing the game competitively as compared with cooperatively or solitarily.

## 4.1. Intergroup behaviour

Our investigation revealed no effects of cooperation and competition on children's subsequent intergroup behaviour. The different contexts of the intervention game did not significantly influence children's in-group bias, as indicated by their sharing and social inclusion. This finding contrasts with the results by Spielman [31], who found that children's in-group favouritism increased after being primed with third-party competition as compared with a neutral or no priming condition. Our data rather support the view that preschoolers' in-group bias cannot be diminished (or fostered) by priming with unrelated interdependent contexts. It appears that the elicitation of a cooperative or competitive psychological orientation (e.g. elicited by a game) do not have spill-over effects on children's prosocial behaviour in third-party intergroup contexts.

Based on his results, Spielman [31] concluded that intergroup competition is an essential element of in-group bias in minimal group situations: only when competition is involved, in-group bias emerges. Our findings do not support this view. Instead, our results resonate with research suggesting that competition between groups is not necessary for children's expectations of between-group harm [16] and in-group bias [14]. A strong interpretation of our data might support views claiming that the mere dichotomous categorization of others into minimal groups can result in in-group bias (for an overview, see [102]). However, when drawing these conclusions, one has to keep in mind that we did not find an in-group bias in our preschool-aged sample (see below).

It has to be noted that we did not replicate Spielman's [31] procedure exactly. Our procedure differed from that of Spielman in three fundamental ways: first, the participants in Spielman's study (6-year-olds) were on average older than the children tested here (4- to 6-year-olds). It might be that older children are

more sensitive to the priming of competitive interdependences. However, this should have been indicated by an interaction between age and condition, which was absent in our data. Second, the gap between the elicitation of the orientation and the assessment of children's sharing was shorter in Spielman's investigation than in the current study. Children distributed the resources immediately after the priming phase in the study by Spielman. In our design, the experimenter and the child went to a separate quiet room which in some day-care centres took a few minutes. Given the subtle nature of priming effects, it might be that we diminished the elicited orientation through this procedural detail. Third, in Spielman's procedure, children have been assigned to a group before the orientation has been elicited, while we did this the other way around. We decided to establish group membership after playing the intervention game to minimize the chance that dyad partners would know their partner's group membership. Here, we wanted the game to be independent of the groups as Spielman concluded that an independent competitive prime can supply a competitive interpretation of an intergroup context. Before the second gaming phase, children were told to keep the groups secret from their dyad partner, and none of the participants mentioned the groups during the gaming phase. It might be that the competition has a more significant impact on children's intergroup behaviour after the establishment of the groups. In particular, the establishment of groups has a cooperative element since children mutually have to agree on these. This collective agreement might diminish the effect of a previously elicited competitive orientation and consequently not affect intergroup behaviour. The mutual agreement might not have the same salience when groups have been established before allowing the competitive orientation to shape the perception of the relation of groups. This interpretation might be an interesting avenue for future research. One could investigate whether a previous group manipulation is a necessary condition for the effects of cooperation and competition found by Spielman [31].

Importantly, our findings do not imply that competition and cooperation have no relevance to intergroup behaviour. Group competition can increase (e.g. [28–30]) and cooperative interactions can reduce in-group bias (e.g. [103]). Our findings suggest that a potential spill-over effect of these contexts does not occur within preschool age. Thus, when aiming to improve intergroup contact in young children one might not simply use cooperation without any relation to the specific groups.

Interestingly, we did not find an impact of group membership on children's sharing and no developmental trend in children's social inclusion behaviour. In other words, there is no indication of an in-group bias in our data. This finding is in contrast with evidence suggesting the emergence of an in-group bias in minimal group contexts around preschool age from similar cultural contexts (e.g. [17,19,23,25,26,94]). Our results are in line with the studies by Spielman [31] and Plötner et al. [24], who found children's sharing to be independent of the recipients' group membership, suggesting that children's sensitivity for conventional groups seems to emerge after preschool age. Notably, all children who were part of the statistical analyses passed a comprehension check and could identify their own and the interactants' group memberships correctly. Only three children did not pass the comprehension check and were excluded from data analyses. Children appear to have a robust capacity to perceive such groups, but the effect of such groups on their prosocial behaviour seems rather fragile. However, as children's prosocial behaviours are poorly related [104,105], one should be careful when claiming that children do not generally show an in-group bias in minimal group contexts. It might be that children's in-group bias is expressed by helping or affiliative behaviours (e.g. liking), which we did not assess in the current study. Also, it might be that our measures were not sensitive enough to detect the effect of the groups. Nevertheless, our null results for children's in-group bias point to the value of studies examining the effects of minimal groups on diverse prosocial behaviours throughout early and middle childhood. Ideally, such an investigation is organized in a collaborative research project involving multiple laboratories conducting the same procedure and thereby generating reliable data (similar to [106–109]). Further, it might be interesting to take a closer look at procedural details (e.g. how exactly group membership is established) to learn more about the effects found in previous studies. Open materials seem a promising solution in this endeavour.

## 4.2. Prosociality

Our findings do not corroborate that playing a game with merely a cooperative, competitive or solitary goal structure influences children's sharing toward third-parties. The total number of shared stickers with an in-group and an out-group member was not affected by the context of the game. This finding contrasts with studies suggesting that elicited cooperative and competitive orientations have an impact on the behaviour toward third-parties (e.g. for children: [53,54,56]; for adults: [110]). Here, we cannot

replicate Toppe *et al.*'s [56] effect on preschoolers' sharing in a larger sample and a more controlled experimental setting. Possibly, interdependent interactions only elicit an orientation that is specific for the actors involved in the interaction. In other words, cooperation between Person A and B might only influence prosociality between these parties, but not toward third-parties, who were not involved in the interaction (e.g. [24,36–42,44–52]).

However, one crucial difference in the procedure of the current study and the study by Toppe *et al.* [56] might be responsible for the different findings. The intervention game introduced in this study was less interactive than the one used by Toppe *et al.* [56] and did not require any coordination between the players. In the current study, children did not directly interact but rather acted parallelly when playing the game cooperatively, while children were forced to coordinate their actions in Toppe *et al.*'s design [56] as they were holding strings for playing with the apparatus. Here, we decided to use a less interactive game to control for children's goal achievement in the game (i.e. winning and losing) and to isolate the effect of goal interdependence from the players' coordination of actions. We find that mere goal interdependence does not influence children's prosocial behaviour toward third-parties. Thus, coordination might be necessary to elicit the spill-over effects of cooperation and competition on prosociality toward third-parties. This conclusion would be in line with studies suggesting that collaboration—a highly coordinated form of cooperation—is related to children's promoted sense of fairness [49,50,111,112]. Further, most of the previous studies finding an effect of cooperation and competition on young children's prosocial behaviour used games demanding coordination between co-players [24,36,37,41,44,46,49,50,52,54,55].

Interdependent interactions have many dimensions, such as the relation of goals, coordination or mutual dependence [113,114], and it might be that the interplay of these dimensions—and not a single dimension *per se*—is crucial for effects on young children's prosocial behaviour. The investigation of how the interplay of different dimensions of interdependence shape children's (pro)social behaviour might be a promising area for future research. It might be that preschoolers require settings with high interdependence on many dimensions to be affected by these since their collective intentionality is still weak [115]. Similar approaches acknowledging diverse dimensions of interdependence have been suggested for the investigation of children's social cognition (e.g. visual perspective-taking; [116]). Social interdependence theory and its predictions are broad [2], and a more detailed approach addressing diverse dimensions of interdependent interactions might be needed to learn more about their effects on children's social behaviour. The influence of interactions with different relations of goals (cooperation versus competition) and degree of coordination (high versus low) on preschoolers' prosocial behaviours might be a fruitful investigation in this endeavour.

## 4.3. Gaming engagement

The only behaviour which was influenced by the context of the intervention game was children's physical engagement while playing the game. However, the effect of cooperative, competitive and solitary context interacted with the gaming phase. Across all conditions, children performed more tube rotations in the second gaming phase. This increase was particularly pronounced in the competitive condition. In other words, a competition promoted children's physical engagement not immediately, but only after a specific duration of play (i.e. in the second gaming phase). This finding agrees with previous work suggesting a promoted performance during competitive encounters [58,59,117]. With the null results for children's sharing and social inclusion in mind, this finding suggests that cooperation and competition reliably influence children's behaviour when being in such contexts but not in subsequent unrelated situations. However, it may also be that the spill-over effects of cooperation and competition on prosociality toward third-parties require a longer gaming duration (i.e. more than two gaming phases). It might be that only repeated gaming experiences in the respective context elicit the hypothesized effects. Importantly, one has to note that the interaction effect between the gaming contexts and the gaming phases on children's physical engagement is explorative as its investigation has been suggested by a reviewer. Thus, a replication of this effect is needed to draw reliable conclusions.

The engagement during the game and its interaction with the gaming context did not influence children's sharing or social inclusion. Again, these null results support the idea that children's interdependent interactions with their peers have a limited impact on their prosociality in subsequent interactions with uninvolved third-parties. However, children's physical engagement is not necessarily equivalent to their psychological engagement. For example, children who are highly engaged in a competition might focus more on their competitor's performance and not put all efforts into their own

physical activity. Future research might assess other behaviours such as children's emotional reactions after a round as these might be more suitable predictors for their psychological engagement.

## 4.4. Social dominance orientation

In a separate analysis, we investigated how parental SDO—the preference for inequality among social groups [60,61]—accentuates the effect of cooperation and competition on children's intergroup behaviour. Our results do not indicate that parents' SDO pronounces the effect of cooperation and competition on children's intergroup behaviour. Also, we do not see a direct effect of parents' SDO on their children's behaviour. This finding disagrees with studies finding a link between parents' and their children's intergroup attitudes (e.g. [73]).

Differences of the assessment might explain the missing link between parents' SDO and children's behaviour. In a meta-analysis, Degner & Dalege [73] found that the conceptual overlap between measures moderates the link between parent–child similarities in intergroup attitudes. In the current study, children were part of an immediate intergroup context, while parents filled out a somewhat unspecific questionnaire (i.e. not related to a particular group). Furthermore, children's data were assessed by observing their behaviour, whereas parents' data were collected through self-report. Thus, the methodological difference in the assessment of children's and parents' intergroup behaviour might have been too large to detect a potential link. The missing connection might also be explained by the dissimilarity of the groups to which the respective assessment referred. In the current study, children were confronted with temporary and minimal groups, while the parents' questionnaire referred to real but unspecific social groups. Using the same kind of groups (e.g. minimal groups) for the parents' and the children's measurement could reveal a possible connection between parents' and their children's intergroup behaviour.

However, the small sample size of the subsample ($n = 54$) generally hinders firm conclusions. Future research on parent–child similarities in intergroup behaviour might use matched assessment methods and groups. For example, children could participate in the current version of the social inclusion paradigm while parents participate in a digital version of this task (e.g. similar to Cyberball paradigms; [118,119]) using minimal groups for both parents' and children's assessment. Matching the assessment might reveal interesting links between parents' and children's intergroup behaviour and the socialization of such behaviours.

Besides, in the current study, we used an explicit measurement of parental intergroup attitudes. This explicit measurement leads to methodological problems such as social desirability (see [62]). Past research suggests that children are more likely to adopt their parents' implicit rather than explicit attitudes [72,120,121]. Thus, the implicit measurement of intergroup attitudes might be a promising approach for future studies.

## 5. Conclusion

In sum, we investigated how cooperative, competitive and solitary orientations influence German 4- to 6-year-olds' prosocial behaviour in an intergroup context. Children played a cooperative, competitive and solitary game. Hereafter, children's sharing and social inclusion were assessed. The cooperative, competitive and solitary gaming context did not affect children's intergroup behaviour. Also, our results question the emergence of an in-group bias in minimal group contexts within preschool age. We could not replicate findings suggesting a general promotive effect of a cooperative orientation on children's prosociality compared with a competitive orientation. Children's physical engagement was more pronounced when playing the game competitively as compared with cooperatively or solitarily. Thus, while being in a cooperative and competitive context, children adjust their behaviour. However, the effects of cooperative and competitive interactions on children's social behaviour in subsequent third-party contexts seem to be weak or even not present.

Ethics. The study was part of a project that has been approved by the ethics committee of the Medical Faculty of Leipzig University (project name: 'Non-pathological development of social behaviours and competences in children and adults with behaviour-based observational, peripheral physiological and psychometrical methods', protocol number: 169/17-ek). All parents signed a written informed consent form for their children.
Data accessibility. Data and script for analyses available from the Open Science Framework: osf.io/pu89t/.

Authors' contributions. T.T., S.H., F.Z. and D.B.M.H. designed the study; T.T. and F.Z. collected data; T.T. and S.H. conducted the statistical analyses; T.T. drafted the manuscript; S.H., F.Z. and D.H. critically revised the manuscript; all authors gave final approval for publication.

Competing interests. We declare we have no competing interests.

Funding. The authors received no specific funding for this work.

Acknowledgements. We are grateful to all the children who participated in the study. We thank Sina Gibhardt, Astrid Seibold and Katja Kirsche, for their assistance for data collection, Ludwig Paeth for reliability coding, as well as Roman Stengelin and Marie Jolanda Kaiser for helpful comments on the study procedure and the manuscript.

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
