## [Peer Review File · Royal Society Open Science]

Review History

RSOS-200494.R0 (Original submission)

Review form: Reviewer 1

Is the manuscript scientifically sound in its present form?

Yes

Are the interpretations and conclusions justified by the results?

No

Is the language acceptable?

Yes

Do you have any ethical concerns with this paper?

No

Have you any concerns about statistical analyses in this paper?

Yes

Recommendation?

Major revision is needed (please make suggestions in comments)

Comments to the Author(s)

See attached file (Appendix A).

Review form: Reviewer 2**Is the manuscript scientifically sound in its present form?**

Yes

Are the interpretations and conclusions justified by the results?

No

Is the language acceptable?

Yes

Do you have any ethical concerns with this paper?

No

Have you any concerns about statistical analyses in this paper?

No

Recommendation?

Reject

Comments to the Author(s)

Review of The Influence of Cooperation and Competition on Preschoolers' Prosociality Toward In-Group and Out-Group Members

Summary

This paper describes results from a study with children investigating the downstream consequences of cooperative, competitive and solitary activities. Specifically, the study tests how these activities influence 4- to 6-year-olds' sharing and inclusion behavior toward in- and out-group members.

General comments

The authors should be applauded for embracing open science methods. Their OSF page was easy to navigate and it is great to see that this study was preregistered. In this same vein, I very much agree with the authors that replications are valuable and so appreciate their goal of providing a conceptual replication of Spielman (2000, PSPB). Additionally, I found their materials and games to be inventive and age-appropriate. Despite these clear strengths, I found the paper to be unconvincing with respect to its main claims and interpretations. I outline my reasons below, along with some relatively less major comments and queries.

Main comments

As the authors note in their discussion, their study design departs from Spielman in what I think is a very important way: namely, the minimal group manipulation was conducted after the cooperation/competition manipulations. The rationale for this change and other changes is explained clearly enough in the discussion (though I think this should also be articulated in the introduction), yet claims and interpretations – particularly about the lack of group membership

effects – are not adjusted accordingly. Moreover, since one of the main goals of this study was to provide a conceptual replication of this prior work, it seems to me that a valuable contribution of this line of work would be to further probe the conditions under which effects do appear. For instance, it would be very interesting to know whether the group manipulation upstream of cooperation/competition is essential and the addition of a direct replication would contribute to a much richer story.

The authors seem to call into question the presence of minimal group effects on sharing based on their null effect. However, no manipulation check was conducted to demonstrate that the minimal group manipulation was successful. It is possible that children did indeed care about their minimal groups yet this had no effect on their sharing. However, it is also possible that the minimal group manipulation did not successfully induce strong ingroup preferences. If the authors had conducted a complementary preference check, perhaps at the very end of the task, their claims about the lack of a group effects on sharing would be much better-grounded. The fact that they saw a group effect in the inclusion task is somewhat helpful in this regard. However, in the absence of a manipulation check, it is difficult to know how strong ingroup preferences were (e.g., maybe a very weak ingroup preference is enough for the inclusion task but not the sharing task) and/or whether some children showed an ingroup preference while others did not which could help account for the sharing results.

As mentioned above, it is great that this study was preregistered. However, there were a few differences between the preregistration and the study that gave me pause. First, the preregistration places a lot of emphasis on questions surrounding Social Dominance Orientation (SDO). However, this is conspicuously absent from the MS. For instance, more than half of the section under “Specify exactly which analyses you will conduct to examine the main question/hypothesis” in the preregistration is devoted to SDO yet this is not mentioned at all in the MS. It seems important, at the very least, to address this discontinuity. Second, the preregistration states that 192 children will be tested but the MS reports that $N = 144$ were tested. Later in the MS, it is clear that 96 children were tested in the solitary condition which would seem to indicate that the N was indeed 192 (48 in cooperative, 48 in competitive, 96 in solitary) which means that there may simply be an error under “design and procedure” but it left me a bit confused as to how many children were actually included. If there is indeed a discrepancy between the MS and the prereg, this should be addressed explicitly. Particularly given that the reported power analysis is associated with an N of 144.

On the subject of the power analysis, it would be helpful to make clear what effect was being tested in the power analysis. Given that the interaction between age and condition was a focus of the MS and preregistration, was this the effect of interest in power analyses? Or was the study powered to detect a main effect of condition? If not the age \times condition interaction, this would need to be addressed by, minimally, labelling these analyses as exploratory. Additionally, how were power analyses run? Were simulations used to accommodate the mixed models, was power approximated using something like g^* power or something else? More details here would be very helpful.

Finally, given that this paper focuses on children, I had several questions about the way age was dealt with. As mentioned above, it was unclear to me if the interaction between age \times condition was the main effect of interest. If so, I was curious about how children were recruited to ensure roughly even spread across age. Or, if recruitment was not directed in this way, what was the spread across ages that organically emerged from your sampling and how might this influence age-based analyses?

Other comments

- I found the introduction and, to a lesser extent the discussion, quite hard to navigate because it jumps around a fair bit in terms of content and organization. The discussion was slightly easier to follow thanks to the subheadings, but I would urge the authors to run through both sections again, making sure that the content is organized in the most streamlined way and smoothing out the grammar (e.g., verb tenses jump around a bit too)
- I had to reread the procedure for the inclusion task several times before understanding the difference between the 'normal' rounds and the directive trial. Specifically, that in the former the puppets spoke out loud but *to themselves* while in the latter they spoke to the child. I now understand this distinction but it might still be worth highlighting this difference very clearly so that it's apparent how the directive trial is different.
- I understand the authors' rationale for not reporting data for the 'baseline' children and agree that this condition does not seem like a clear baseline. However, it would still be worth reporting their data in the interest of transparency. Their post-hoc exclusion made me wonder whether their data were hard to explain for some reason.
- It would be helpful if the authors could be clear about when they are referring to dyads vs participants (e.g., N = 48 participants, N = 24 dyads). I think this might help clear up some general confusion about the sample size.
- Since the authors already employ a partial Bayesian approach and are thus familiar with Bayesian methods, I wondered if they considered running their models in a Bayesian framework. Of course, I appreciate that this was not the preregistered approach and certainly wouldn't advocate for replacing the current models. However, since the authors make claims that hinge on null findings, a Bayesian approach would allow them to assign a probability to the null hypothesis.

Decision letter (RSOS-200494.R0)

Dear Mr Toppe:

Manuscript ID RSOS-200494 entitled "The Influence of Cooperation and Competition on Preschoolers' Prosociality Toward In-Group and Out-Group Members" which you submitted to Royal Society Open Science, has been reviewed. The comments from reviewers are included at the bottom of this letter.

In view of the criticisms of the reviewers, the manuscript has been rejected in its current form. However, a new manuscript may be submitted which takes into consideration these comments.

Please note that resubmitting your manuscript does not guarantee eventual acceptance, and that your resubmission will be subject to peer review before a decision is made.

Your resubmitted manuscript should be submitted by 23-Dec-2020. If you are unable to submit by this date please contact the Editorial Office.

Reviewers' Comments to Author:

Reviewer: 1

Comments to the Author(s)
see attached file

Reviewer: 2

Comments to the Author(s)

Review of The Influence of Cooperation and Competition on Preschoolers' Prosociality Toward In-Group and Out-Group Members

Summary

This paper describes results from a study with children investigating the downstream consequences of cooperative, competitive and solitary activities. Specifically, the study tests how these activities influence 4- to 6-year-olds' sharing and inclusion behavior toward in- and out-group members.

General comments

The authors should be applauded for embracing open science methods. Their OSF page was easy to navigate and it is great to see that this study was preregistered. In this same vein, I very much agree with the authors that replications are valuable and so appreciate their goal of providing a conceptual replication of Spielman (2000, PSPB). Additionally, I found their materials and games to be inventive and age-appropriate. Despite these clear strengths, I found the paper to be unconvincing with respect to its main claims and interpretations. I outline my reasons below, along with some relatively less major comments and queries.

Main comments

As the authors note in their discussion, their study design departs from Spielman in what I think is a very important way: namely, the minimal group manipulation was conducted after the cooperation/competition manipulations. The rationale for this change and other changes is explained clearly enough in the discussion (though I think this should also be articulated in the introduction), yet claims and interpretations – particularly about the lack of group membership effects – are not adjusted accordingly. Moreover, since one of the main goals of this study was to provide a conceptual replication of this prior work, it seems to me that a valuable contribution of this line of work would be to further probe the conditions under which effects do appear. For instance, it would be very interesting to know whether the group manipulation upstream of cooperation/competition is essential and the addition of a direct replication would contribute to a much richer story.

The authors seem to call into question the presence of minimal group effects on sharing based on their null effect. However, no manipulation check was conducted to demonstrate that the minimal group manipulation was successful. It is possible that children did indeed care about

their minimal groups yet this had no effect on their sharing. However, it is also possible that the minimal group manipulation did not successfully induce strong ingroup preferences. If the authors had conducted a complementary preference check, perhaps at the very end of the task, their claims about the lack of a group effects on sharing would be much better-grounded. The fact that they saw a group effect in the inclusion task is somewhat helpful in this regard. However, in the absence of a manipulation check, it is difficult to know how strong ingroup preferences were (e.g., maybe a very weak ingroup preference is enough for the inclusion task but not the sharing task) and/or whether some children showed an ingroup preference while others did not which could help account for the sharing results.

As mentioned above, it is great that this study was preregistered. However, there were a few differences between the preregistration and the study that gave me pause. First, the preregistration places a lot of emphasis on questions surrounding Social Dominance Orientation (SDO). However, this is conspicuously absent from the MS. For instance, more than half of the section under "Specify exactly which analyses you will conduct to examine the main question/hypothesis" in the preregistration is devoted to SDO yet this is not mentioned at all in the MS. It seems important, at the very least, to address this discontinuity. Second, the preregistration states that 192 children will be tested but the MS reports that $N = 144$ were tested. Later in the MS, it is clear that 96 children were tested in the solitary condition which would seem to indicate that the N was indeed 192 (48 in cooperative, 48 in competitive, 96 in solitary) which means that there may simply be an error under "design and procedure" but it left me a bit confused as to how many children were actually included. If there is indeed a discrepancy between the MS and the prereg, this should be addressed explicitly. Particularly given that the reported power analysis is associated with an N of 144.

On the subject of the power analysis, it would be helpful to make clear what effect was being tested in the power analysis. Given that the interaction between age and condition was a focus of the MS and preregistration, was this the effect of interest in power analyses? Or was the study powered to detect a main effect of condition? If not the age \times condition interaction, this would need to be addressed by, minimally, labelling these analyses as exploratory. Additionally, how were power analyses run? Were simulations used to accommodate the mixed models, was power approximated using something like *g*power* or something else? More details here would be very helpful.

Finally, given that this paper focuses on children, I had several questions about the way age was dealt with. As mentioned above, it was unclear to me if the interaction between age \times condition was the main effect of interest. If so, I was curious about how children were recruited to ensure roughly even spread across age. Or, if recruitment was not directed in this way, what was the spread across ages that organically emerged from your sampling and how might this influence age-based analyses?

Other comments

- I found the introduction and, to a lesser extent the discussion, quite hard to navigate because it jumps around a fair bit in terms of content and organization. The discussion was slightly easier to follow thanks to the subheadings, but I would urge the authors to run through both sections again, making sure that the content is organized in the most streamlined way and smoothing out the grammar (e.g., verb tenses jump around a bit too)

- I had to reread the procedure for the inclusion task several times before understanding the difference between the 'normal' rounds and the directive trial. Specifically, that in the former the puppets spoke out loud but *to themselves* while in the latter they spoke to the child. I now understand this distinction but it might still be worth highlighting this difference very clearly so that it's apparent how the directive trial is different.

- I understand the authors' rationale for not reporting data for the 'baseline' children and agree that this condition does not seem like a clear baseline. However, it would still be worth reporting their data in the interest of transparency. Their post-hoc exclusion made me wonder whether their data were hard to explain for some reason.

- It would be helpful if the authors could be clear about when they are referring to dyads vs participants (e.g., N = 48 participants, N = 24 dyads). I think this might help clear up some general confusion about the sample size.

- Since the authors already employ a partial Bayesian approach and are thus familiar with Bayesian methods, I wondered if they considered running their models in a Bayesian framework. Of course, I appreciate that this was not the preregistered approach and certainly wouldn't advocate for replacing the current models. However, since the authors make claims that hinge on null findings, a Bayesian approach would allow them to assign a probability to the null hypothesis.

Author's Response to Decision Letter for (RSOS-200494.R0)

See Appendix B.

RSOS-202171.R0

Review form: Reviewer 1

Is the manuscript scientifically sound in its present form?

Yes

Are the interpretations and conclusions justified by the results?

Yes

Is the language acceptable?

Yes

Do you have any ethical concerns with this paper?

No

Have you any concerns about statistical analyses in this paper?

No

Recommendation?

Accept as is

Comments to the Author(s)

I was a reviewer for the original submission. The revision was a pleasure to read, and I appreciate the authors' clear efforts to address the reviewers' concerns. Great job on this work!

Review form: Reviewer 2

Is the manuscript scientifically sound in its present form?

Yes

Are the interpretations and conclusions justified by the results?

Yes

Is the language acceptable?

Yes

Do you have any ethical concerns with this paper?

No

Have you any concerns about statistical analyses in this paper?

No

Recommendation?

Accept as is

Comments to the Author(s)

This was a very responsive revision and I appreciate the amount of work that went into addressing all the comments. I think the revised paper is much stronger thanks to the changes the authors made. My only suggestion – which the authors can take or leave – is to slightly revise the section on why no ingroup bias effects were seen. My previous comment was suggesting that the group manipulation itself may have failed. If this were the case it's no surprise that bias was not seen in the measures at hand. Comp checks alone are not sufficient to show that children *cared* about their groups. Group manipulation checks (e.g., asking some preference questions) are now fairly standard in this kind of work and could have been helpful here. I don't think there's much to be done about this except that manipulation checks like this may be helpful in future lines of work that follow on from this and it may be worth engaging with this issue a bit in the discussion. The section added to the revised MS gets at this a bit but doesn't tackle the possibility that the group manipulation may not have worked more directly.

Decision letter (RSOS-202171.R0)

Dear Mr Toppe,

I am pleased to inform you that your manuscript entitled "The Influence of Cooperation and Competition on Preschoolers' Prosociality Toward In-Group and Out-Group Members" is now accepted for publication in Royal Society Open Science.

Please ensure that you send to the editorial office an editable version of your accepted manuscript, and individual files for each figure and table included in your manuscript. You can send these in a zip folder if more convenient. Failure to provide these files may delay the

processing of your proof. You may disregard this request if you have already provided these files to the editorial office.

on behalf of Essi Viding (Subject Editor)
openscience@royalsociety.org

Reviewer comments to Author:
Reviewer: 1

Comments to the Author(s)

I was a reviewer for the original submission. The revision was a pleasure to read, and I appreciate the authors' clear efforts to address the reviewers' concerns. Great job on this work!

Reviewer: 2

Comments to the Author(s)

This was a very responsive revision and I appreciate the amount of work that went into addressing all the comments. I think the revised paper is much stronger thanks to the changes the authors made. My only suggestion – which the authors can take or leave – is to slightly revise the section on why no ingroup bias effects were seen. My previous comment was suggesting that the group manipulation itself may have failed. If this were the case it's no surprise that bias was not seen in the measures at hand. Comp checks alone are not sufficient to show that children *cared* about their groups. Group manipulation checks (e.g., asking some preference questions) are now fairly standard in this kind of work and could have been helpful here. I don't think there's much

to be done about this except that manipulation checks like this may be helpful in future lines of work that follow on from this and it may be worth engaging with this issue a bit in the discussion. The section added to the revised MS gets at this a bit but doesn't tackle the possibility that the group manipulation may not have worked more directly.

Appendix A

In this manuscript, Toppe and colleagues test the priming effects of different gaming contexts (cooperation, competition, solitary) in young children's (4- to 6-year-olds) sharing behaviors with and inclusion of ingroup/outgroup members. Phases of the experiment were completed in the following order: (1) the intervention game (cooperation, competition, or solitary), establishing the context, (2) group assignment, (3) one set of participants (P1) played 2 dictator games, one with an ingroup member and one with an outgroup member; the second set of participants (P2) played a social-inclusion game, (4) the intervention game again, (5) P2 played the dictator games, P1 played the inclusion game. The authors found that gaming context did not affect children's intergroup behavior or sharing behavior. They did, however, find some evidence of greater ingroup bias with increasing age. This study is ambitious and tries to address the mixed evidence found in the literature.

My main concerns revolve around the interpretation of the null effects.

1) May this study be underpowered? In the Methods section, the authors state that the power analysis was based on a medium effect of .80, but it's unclear what type of effect it refers to (interaction effect? main effect?). I think it might be worth conducting a sensitivity analysis to instead assess the smallest effect size this study design has 80% power to detect.

2) On a different note, I wonder how the different gaming phases may have affected any experimental effects. That is, one possible effect of the second phase of the intervention game is that it reinforced the gaming context. If so, we may expect to see greater effects of gaming context in the dictator game and social inclusion game in the second phase than the first phase. Is that the case? Is there an interaction between condition and gaming phase?

3) How were dyads determined? Would factors like familiarity with the other person affect the effectiveness of the gaming context?

4) Did the authors have any way to assess whether the minimal group manipulation was effective? Right now, it's unclear whether the minimal group manipulation 'worked' at all. In the best-case scenario, participants *did* display an ingroup bias but just not for sharing or inclusion behaviors. Is there any evidence of participants showing an ingroup bias in explicit preference or liking?

I applaud the authors for providing clear limitations to the study in the Discussion and encourage them to provide greater discussion of further limitations raised in the above points.

Minor comments:

- In the introduction, the authors describe prior work motivating this study as manipulating gaming context to prime cooperative orientation, etc. The intention to use gaming context as a prime got lost in the manuscript but should really be emphasized.

Otherwise, it's difficult to wrap one's head around a theoretical justification for why we should see an effect of condition in sharing and inclusion when the condition is manipulated in a completely separate task.

- Can the authors provide the breakdown of the sample for each age group (4, 5, 6 year-olds)? There are many developmental differences between 4-year-olds and 6-year-olds.
- The reasoning behind running two phases of the intervention game was a bit unclear. Did the authors do this to strengthen the possible effect of gaming context? Or did they only do this so that people who drew in the solitary condition could now get the chance to play the game? An explicit mention of the reasoning behind this decision would help the reader understand why this was done in two phases.
- In Table 1, for all cases, we see above-chance giving to outgroup members. Why might this be the case? (Is it actually significantly above 50%?). This is interesting because it suggests that people may even have an outgroup-bias.
- The labels for Figure 2 are too fuzzy to read. Also, for Figure 2b, it might be useful to include a key that describes what the dark and light circles denote (as opposed to noting it in just the caption).
- In the pre-registration, there were hypotheses involving parental social dominance orientation. Can the authors either incorporate these analyses in the paper or explicitly say why they decided to stray away from the pre-registration? Generally, it might be helpful to the reader to explicitly point out all deviations from the pre-registration and reasoning for the deviation.

OSF-related:

- I appreciate the inclusion of the procedure on OSF, but I think the current version is a bit confusing to go through. I think it would be easier for the reader to access the procedure if the German text were separated from the English text (right now, it's intermixed).
- A data dictionary would help the reader interpret what each column represents (e.g., what's Trial 1? Trial 1 of which task, etc.)

Response letter

Dear Andrew Dunn,

Thank you for your response and the opportunity to revise our manuscript (*RSOS-200494*). The suggestions offered by you and the reviewers have been immensely helpful. We have included the Reviewers' comments and responded to each comment individually, indicating how we addressed each concern and describing the changes we have made. On the next pages, you find our responses to all reviewers' comments.

Amongst others, both reviewers gave notations to our excluded survey data on parents' social dominance orientation. We included this data into our analyses and revised the Introduction and Discussion.

In response to Reviewer 1's comment, we added the interaction between the gaming phase and the experimental condition into our analyses. This did not change the broad pattern of the result, such that the cooperative and competitive context of the intervention game did not influence children's sharing and social inclusion. However, a new finding is that children's physical effort while playing the game was impacted by the interaction between the experimental condition and the gaming phase. The new (explorative) result suggests that children's physical effort increased across gaming phases but that this increase was the strongest in the competitive as compared to the cooperative and solitary condition.

To improve reproducibility, we uploaded a video of the procedure of the inclusion task with an adult participant.

We hope the revised manuscript will better suit the requirements of *Royal Society Open Science*. We are happy to consider further revisions, and we thank you for your continued interest in our research.

Sincerely,

Theo Toppe
Max Planck Institute for Evolutionary Anthropology
Department of Comparative Cultural Psychology
Deutscher Platz 6, 04103 Leipzig
E-mail: theo_toppe@eva.mpg.de

Susanne Hardecker
SRH University of Applied Health Sciences Gera

Franca Zerres
Leipzig University

Daniel B. M. Haun
Max Planck Institute for Evolutionary Anthropology

Reviewer 1

Comment 1:

May this study be underpowered? In the Methods section, the authors state that the power analysis was based on a medium effect of .80, but it's unclear what type of effect it refers to (interaction effect? main effect?). I think it might be worth conducting a sensitivity analysis to instead assess the smallest effect size this study design has 80% power to detect.

Response:

We thank Reviewer 1 for this comment. The power analysis referred to the overall effect of the predictor variables (i.e., the interaction of age and condition and main effects of condition, age, and engagement). We added this information to the methods section:

"The sample size of $N = 144$ was suggested by a prior power analysis expecting a medium effect with a statistical power of .80 and a probability of .05 for a type I error. This power analysis referred to the overall effect of the variables of interest—interaction between condition and age and the main effects of condition, age, and engagement. The power analysis was conducted with the pwr package [83] in R statistical software ([84]; see R script at osf.io/pu89t/)." (p. 5)

Also, we thank Reviewer 1's suggestion for a sensitivity analysis. It might be that we misunderstood the term "sensitivity analysis" in Reviewer 1's comment. From our view, Reviewer 1 suggested a post-hoc power analysis. However, we do not see a considerable merit in such a post-hoc power analysis since these are assumed to be "tautological and uninformative" (Goodman & Berlin, 1994, p. 202; see also Gelman, 2019; Levine & Ensom, 2001; O'Keefe, 2007). However, it might be that we did not understand Reviewer 1's suggestion correctly. We highly appreciate a clarification of the term "sensitivity analysis" and specific tools (e.g., R packages) to conduct such an analysis for generalized linear mixed models.

With respect to this comment, we also want to mention that we calculated Bayes factors in the revised manuscript as these were suggested by Reviewer 2. These Bayes factors offer additional information about the detected effects and null results. The critical part of the manuscript reads as:

"Further, we calculated Bayes factors (BFs) comparing the evidence for the full model with the evidence for the null model given our data. Bayes factors are a numerical value quantifying how well a hypothesis predicts the given data relative to a competing hypothesis [98]. Here, we used the bayestestR package [99] to extract the Bayes factor for the full and the null model and applied Jeffrey's system [100] to interpret these. The advantage of Bayes factors is that they allow for a differentiated interpretation of the support of the null versus the alternative hypothesis." (p. 9)

Gelman (2019). Don't Calculate Post-Hoc Power Using Observed Estimate of Effect Size. *Annals of Surgery*, 269(1), e9.

Goodman & Berlin (1994). The Use of Predicted Confidence Intervals When Planning Experiments and the Misuse of Power When Interpreting Results. *Annals of Internal Medicine*, 121(3), 200–206.

Levine & Ensom (2001). Post Hoc Power Analysis: An Idea Whose Time Has Passed?" *Pharmacotherapy*, 21(4): 405–9.

O'Keefe (2007). Brief Report: Post Hoc Power, Observed Power, A Priori Power, Retrospective Power, Prospective Power, Achieved Power: Sorting Out Appropriate Uses of Statistical Power Analyses. *Communication Methods and Measures*, 1(4), 291–99.

Comment 2:

On a different note, I wonder how the different gaming phases may have affected any experimental effects. That is, one possible effect of the second phase of the intervention game is that it reinforced the gaming context. If so, we may expect to see greater effects of gaming context in the dictator game and social inclusion game in the second phase than the first phase. Is that the case? Is there an interaction between condition and gaming phase?

Response:

We highly acknowledge Reviewer 1's argument and included the interaction between condition and gaming phase into the analysis. To counter the increased type-I error probability caused by the addition of this new interaction term, we slightly changed our statistical approach. In all analyses, we compared a full model including all variables of interest (predictors and controls) with a null model merely comprising control variables. Only if the likelihood ratio test between the full and the null model

was significant, we continued the statistical analysis of the full model with the same approach outlined in the original version of the manuscript.

Following this more conservative approach, our results changed minorly. Again, we did not find any effects of the variable of interest for children's sharing and social inclusion. In contrast to the previous analysis, the interaction between condition and gaming phase influenced children's physical engagement (i.e., their tube rotations), such that children playing a competitive game showed a stronger increase over the two gaming phases as compared to the cooperative and solitary condition.

We revised the methods, results, and discussion section in the manuscript accordingly. The critical passages read as follows:

"In addition to our preregistration, we also explored whether the duration of playing the game moderated the potential effect of the different gaming contexts. It might be that the expected effects of the gaming contexts become more pronounced after playing the game for a while as children's reactions to the contexts might be delayed. The idea for this explorative analysis occurred in the review process and was integrated into our preregistered analysis." (p. 5)

"To statistically test our hypotheses, we fitted generalized linear mixed models (GLMMs; [95]) in R statistical software [84] using the lme4 package [96]. For all models, we compared a full model comprising all predictors of interest and control variables with a null model comprising control variables only. Only in case of a significant full-null model comparison, we tested the effect of individual predictors by comparing the full model with reduced models lacking them one at a time. By such full-null model comparisons multiple testing can be avoided [97]. In case of non-significant interactions, we continued our analysis with a model comprising the related main effects only. For all model comparisons, we conducted likelihood ratio tests." (p. 9)

Comment 3:

How were dyads determined? Would factors like familiarity with the other person affect the effectiveness of the gaming context?

Response:

The information influencing dyad constellation were children's sex and age, such that dyads partners had the same sex and needed to be of approximately the same age (maximal difference six months). As children participated in their daycare centers, children often knew their co-players. However, we did not document the relationship between co-players and did not have any prior prediction on how this might influence the effect of the experimental conditions. It might be that friendship fosters the effect of cooperation and buffers the effect of competition.

However, the moderating effect of players' relationships on the impact of cooperation and competition is beyond the scope of this manuscript and needs more targeted approaches. Also, it seems hard to discuss potential effects of players' relationships as our data is not comprehensive in this respect.

Nevertheless, we agree that this aspect should be mentioned in the manuscript. We included information about dyad determination into the Participants section:

"Children participated in same sex dyads. In each dyad, children were aged equally such that the maximum difference between children's age was not larger than 6 months. Besides the criteria of age and sex, dyads were determined randomly. As children were tested in their day-care centre, they were often familiar with their co-player but dyad constellation was not influenced by children's relationship with each other." (p. 5)

Comment 4:

Did the authors have any way to assess whether the minimal group manipulation was effective? Right now, it's unclear whether the minimal group manipulation 'worked' at all. In the best-case scenario, participants did display an ingroup bias but just not for sharing or inclusion behaviors. Is there any evidence of participants showing an ingroup bias in explicit preference or liking?

Response:

We thank Reviewer 1 for this comment. In our study, we do not find any sign for children's in-group bias. To ensure that children understood the minimal group situation, we conducted a comprehension check and children had to state their own group membership and whether they shared group membership with the portrayed children (for the dictator game) and the puppets (for the inclusion task). For the dictator game, three children did not pass this comprehension check and were excluded from the respective analysis. For the inclusion task, all children passed the comprehension check. Thus, all data included in the statistical analyses are from children who understood that there were in different groups and could identify their own and their interactants group membership. Accordingly, the necessary conditions for the detection of an in-group bias are fulfilled. Also, our procedure was very similar (or even identical) to previous work which detected in-group biases in children (e.g., Toppe et al., 2020; Dunham et al., 2011).

Given that evidence on preschool-aged children's in-group bias is somewhat mixed and that in-group bias seems to develop throughout preschool age (Over, 2018), our null-result seems plausible.

However, we fully acknowledge that our evidence is not conclusive in this point. Particularly, in early childhood research where minor changes in procedures can affect results substantially (e.g., for minimal group manipulations, see Misch & Dunham, 2021). Therefore, we included the argument in the Discussion. The critical passage reads as follows:

“Interestingly, we did not find an impact of group membership on children’s sharing and no developmental trend in children’s social inclusion behaviour. In other words, there is no indication of an in-group bias in our data. This finding is in contrast to evidence suggesting the emergence of an in-group bias in minimal group contexts around preschool age from similar cultural contexts (e.g., [17,19,23,25,26,94]). Our results are in line with the studies by Spielman [31] and Plötner et al. [24], who found children’s sharing to be independent of the recipients’ group membership, suggesting that children’s sensitivity for conventional groups seems to emerge after preschool age. Notably, all children who were part of the statistical analyses passed a comprehension check and could identify their own and the interactants’ group memberships correctly. Only three children did not pass the comprehension check and were excluded from data analyses. Children appear to have a robust capacity to perceive such groups, but the effect of such groups on their prosocial behaviour seems rather fragile. However, as children’s prosocial behaviours are poorly related [104,105], one should be careful when claiming that children do not generally show an in-group bias in minimal group contexts. It might be that children’s in-group bias is expressed by helping or affiliative behaviours (e.g., liking), which we did not assess in the current study. Also, it might be that our measures were not sensitive enough to detect the effect of the groups. Nevertheless, our null results for children’s in-group bias point to the value of studies examining the effects of minimal groups on diverse prosocial behaviours throughout early and middle childhood. Ideally, such an investigation is organized in a collaborative research project involving multiple laboratories conducting the same procedure and thereby generating reliable data (similar to [106–109]). Further, it might be interesting to take a closer look to procedural details (e.g., how exactly is group membership established) to learn more about the effects found in previous studies. Open materials seem a promising solution in this endeavour.” (p. 11-12)

Dunham, Baron, & Carey (2011). Consequences of 'Minimal' Group Affiliations in Children. *Child Development*, 82(3), 793–811.

Misch & Dunham (2021). (Peer) Group Influence on Children's Prosocial and Antisocial Behavior. *Journal of Experimental Child Psychology*, 201: 104994.

Misch & Dunham (2021). (Peer) Group Influence on Children's Prosocial and Antisocial Behavior. *Journal of Experimental Child Psychology*, 201: 104994.

Over (2018). The Influence of Group Membership on Young Children's Prosocial Behaviour. *Current Opinion in Psychology*, Early Development of prosocial behavior, 20: 17–20.

Toppe, Hardecker, & Haun (2020). Social Inclusion Increases over Early Childhood and Is Influenced by Others' Group Membership. *Developmental Psychology*, 56(2), 324–35.

Comment 5:

I applaud the authors for providing clear limitations to the study in the Discussion and encourage them to provide greater discussion of further limitations raised in the above points.

Response:

We revised the Discussion and aimed to discuss the study's limitations more thoroughly.

Comment 6:

In the introduction, the authors describe prior work motivating this study as manipulating gaming context to prime cooperative orientation, etc. The intention to use gaming context as a prime got lost in the manuscript but should really be emphasized. Otherwise, it's difficult to wrap one's head around a theoretical justification for why we should see an effect of condition in sharing and inclusion when the condition is manipulated in a completely separate task.

Response:

We agree with reviewer 1's comment and revised the introduction. A critical section of interest reads as follows:

"In sum, past research suggests that cooperation and competition can elicit psychological orientations influencing children's in-group bias and prosocial behaviour. The elicitation of a cooperative orientation (e.g., through priming or playing a game) might be a promising intervention on both children's intergroup and prosocial behaviour. On the one hand, a cooperative orientation might reduce preschoolers' in-group bias. On the other hand, it might promote preschoolers' general prosociality toward others. The current study aimed to examine these two effects and to replicate the findings of Spielman [31] and Toppe et al. [56]. Like Toppe et al. [56], we used a dyadic game (after this referred to as intervention game) to elicit a cooperative, competitive, and solitary orientation. After playing the intervention game, we assessed 4- to 6-year-old children's sharing and social inclusion behaviour in a minimal group situation." (p. 3)

Comment 7:

Can the authors provide the breakdown of the sample for each age group (4, 5, 6 year-olds)? There are many developmental differences between 4-year-olds and 6-year-olds.

Response:

In the revised manuscript, we added a written breakdown of the sample for each age group. Further, we offer a histogram of children's age in the supplemental material. The Participants section now reads:

"The sample used for analysis consisted of 144 German children aged between 4 and 6 years (50% female; mean age = 4.96 years; age range = 4.03 to 6.05 years). Children were from a mid-sized German city, and recruitment was based on a laboratory-maintained database, including children from about 150 day-care centres. Participants tested in this study were from 20 day-care centres located in different districts of the city, allowing the assumption that children had diverse socio-economic backgrounds. We aimed to include age as a continuous variable into the statistical analysis and, thus, wanted children's age to be distributed evenly. To achieve a relative even distribution of age across conditions, about one half of the sample should be aged between 4 and 5 and the other half aged between 5 and 6 within each condition ($n_{4y/o} = 78$, $n_{5y/o} = 63$, $n_{6y/o} = 3$; for a histogram of children's age, see supplementary material figure S1). (p. 5)

Comment 8:

The reasoning behind running two phases of the intervention game was a bit unclear. Did the authors do this to strengthen the possible effect of gaming context? Or did they only do this so that people who drew in the solitary condition could now get the chance to play the game? An explicit mention of the reasoning behind this decision would help the reader understand why this was done in two phases.

Response:

Thank you for pointing out this unclarity. We stressed the reasons for this procedure in the revised version of the manuscript. The critical phrase now read as follows:

"After the first assessment of children's sharing and social inclusion, children gave their caps back to the experimenters and reconvened in the test room. Again, children played the intervention game with the same rules as in the first gaming phase after E1 shortly brushed up the rules. In the cooperative and competitive condition, children changed tubes. In the solitary condition, children's roles remained the same, and the target child played the intervention game a second time. Children played the game

in a second phase to minimize the interference of the assessment of the prosocial behaviours and to strengthen the impact of the different contexts of the intervention game. Also, without a second gaming phase the effect of the intervention game might be weaker for the latter assessed prosocial behaviour (for a similar procedure, see Toppe et al. [56]).” (p. 8)

Comment 9:

In Table 1, for all cases, we see above-chance giving to outgroup members. Why might this be the case? (Is it actually significantly above 50%?). This is interesting because it suggests that people may even have an outgroup-bias.

Response:

Inspired by this comment, we calculated a one-sample t-test against an expected value of $\mu = 0.5$ (i.e., inclusion rate above 50%) for children’s inclusion and their decision in the directive trial. For both measures, we found a significant difference between a rate of 50% and the mean inclusion while passing (for the total sample: $M = 0.599$; $t(141) = 2.388$, $p = .018$) and the inclusion in the directive trial respectively (for the total sample: $M = 0.592$; $t(141) = 2.212$, $p = .029$). That is, children’s likelihood to include the out-group puppet was above chance.

Nevertheless, we refrained from including these additional t-tests into the manuscript as these findings are hard to interpret. They might indicate an “out-group bias” as children were highly motivated to include the out-group puppet. However, only by the comparison of a baseline condition without group membership one can interpret the effect of the group constellation. For example, if the inclusion rates in a baseline condition would be equal to the inclusion rates observed in this study, one could not conclude that children have an out-group bias. In such a case, the conclusion would be that group do not affect children’s inclusion rate in this task.

In this respect, we recommend a study by Toppe, Hardecker, and Haun (2020) who directly compared different intergroup constellations in the inclusion task used here. From our view, the t-tests show that children generally have a high willingness to include others into their interactions. The importance of groups for this behavior cannot be determined with our data. Our study aimed at understanding the differences between cooperation and competition on this behavior.

Toppe, Hardecker, & Haun (2020). Social Inclusion Increases Over Early Childhood and Is Influenced by Others’ Group Membership. *Developmental Psychology*, 56(2), 324–35.

Comment 10:

The labels for Figure 2 are too fuzzy to read. Also, for Figure 2b, it might be useful to include a key that describes what the dark and light circles denote (as opposed to noting it in just the caption).

Response:

We excluded Fig 2 due to its unclarity and our new statistical approach (see Comment 2). We decided to describe the findings for children’s sharing in a table (see Table 2, in the manuscript).

Comment 11:

In the pre-registration, there were hypotheses involving parental social dominance orientation. Can the authors either incorporate these analyses in the paper or explicitly say why they decided to stray away from the pre-registration? Generally, it might be helpful to the reader to explicitly point out all deviations from the pre-registration and reasoning for the deviation.

Response:

We included the data of the parental survey on social dominance orientation into the manuscript. This changed different parts of the manuscript. With respect to the deviations to the preregistration, the critical part of the manuscript now reads as:

“First, children would show an in-group bias such that they share more with an in-group than with an out-group member across all experimental conditions. We further investigated how the different gaming contexts (cooperative, competitive, and solitary) would shape the differences between the

stickers shared with the in- and the out-group member and their social inclusion behaviour toward an out-group member. Second, children's total number of shared stickers would be influenced by the different gaming contexts, with more shared stickers after playing a cooperative game as compared to a competitive or a solitary game. Playing a competitive game compared to a cooperative or a solitary game would lead to fewer shared stickers. Third, children's engagement while playing the intervention game would be higher in the competitive context as compared to the cooperative and solitary context. We further explored how children's engagement would moderate their in-group bias and general prosociality. Fourth, parents' SDO would be positively related with in-group bias for children's sharing and negatively related to the general willingness and speed of children's inclusion of an out-group member.

For all hypotheses, we investigated in how far children's age accentuates the effect of the experimental conditions by considering the interaction between condition and age. The consideration of this interaction term seems highly relevant as the understanding of cooperative and competitive contexts seems to emerge throughout preschool age [78–82].

Initially, we planned to test the impact of the contexts of the intervention game, age, engagement, and parents' SDO on each dependent variable in one single model. However, the parental survey response rate was relatively low (42%; $n = 54$; see Materials and Methods). Therefore, we decided to deviate from our preregistration and investigated our fourth hypothesis in a separate data analysis with the respective subsample (see Data analysis). We decided for this deviation to ensure statistical power for the effects of the other predictors.

In addition to our preregistration, we also explored whether the duration of playing the game moderated the potential effect of the different gaming contexts. It might be that the expected effects of the gaming contexts become more pronounced after playing the game for a while as children's reactions to the contexts might be delayed. The idea for this explorative analysis occurred in the review process and was integrated into our preregistered analysis.” (p. 4-5)

Comment 12:

I appreciate the inclusion of the procedure on OSF, but I think the current version is a bit confusing to go through. I think it would be easier for the reader to access the procedure if the German text were separated from the English text (right now, it's intermixed).

Response:

We think that this is a good suggestion and added a fully English version of the procedure (see osf.io/pu89t/). Also, we refer to this detailed description of the procedure in the manuscript (see p. 6).

Comment 13:

A data dictionary would help the reader interpret what each column represents (e.g., what's Trial 1? Trial 1 of which task, etc.)

Response:

We think that this is a good suggestion and added a description of all variables to the R script (see osf.io/pu89t/).

Reviewer 2

Comment 1:

As the authors note in their discussion, their study design departs from Spielman in what I think is a very important way: namely, the minimal group manipulation was conducted after the cooperation/competition manipulations. The rationale for this change and other changes is explained clearly enough in the discussion (though I think this should also be articulated in the introduction), yet claims and interpretations—particularly about the lack of group membership effects—are not adjusted accordingly. Moreover, since one of the main goals of this study was to provide a conceptual replication of this prior work, it seems to me that a valuable contribution of this line of work would be to further probe the conditions under which effects do appear. For instance, it would be very interesting to know whether the group manipulation upstream of cooperation/competition is essential and the addition of a direct replication would contribute to a much richer story.

Response:

We thank Reviewer 2 for this comment and included the differences between the procedure of Spielman's and our study into the Introduction. The critical part of the Introduction reads as follows:

"Spielman [31] did not systematically distinguish the effect of a cooperative orientation on children's prosociality. So far, we do not know whether the effects of a cooperative and a solitary context on preschoolers' in-group bias are similar or not. A systematic investigation of all three contexts—cooperative, competitive, and solitary—is needed to evaluate Spielman's effects. Notably, we slightly changed the order of Spielman's procedure in the current study. In Spielman's study, children have been assigned to a group before the orientation has been elicited. Children were first assigned to a minimal group, then primed with the story, and finally distributed stickers. In the current study, group membership was established after playing the intervention game to minimize the chance that dyad partners would know their partner's group membership. Also, we wanted the game to be independent of the groups to learn more about the impact of cooperative and competitive orientations on novel subsequent intergroup contexts." (p. 4)

Also, we mention this aspect in the Discussion:

"It has to be noted that we did not replicate Spielman's [31] procedure exactly. Our procedure differed from that of Spielman in three fundamental ways: First, the participants in Spielman's study (6-year-olds) were on average older than the children tested here (4- to 6-year-olds). It might be that older children are more sensitive to the priming of competitive interdependences. However, this should have been indicated by an interaction between age and condition, which was absent in our data. Second, the gap between the elicitation of the orientation and the assessment of children's sharing was shorter in Spielman's investigation than in the current study. Children distributed the resources immediately after the priming phase in the study by Spielman. In our design, the experimenter and the child went to a separate quiet room which in some day-care centres took a few minutes. Given the subtle nature of priming effects, it might be that we diminished the elicited orientation through this procedural detail. Third, in Spielman's procedure, children have been assigned to a group before the orientation has been elicited, while we did this the other way around. We decided to establish group membership after playing the intervention game to minimize the chance that dyad partners would know their partner's group membership. Here, we wanted the game to be independent of the groups as Spielman concluded that an independent competitive prime can supply a competitive interpretation of an intergroup context. Before the second gaming phase, children were told to keep the groups secret from their dyad partner, and none of the participants mentioned the groups during the gaming phase. It might be that the competition has a more significant impact on children's intergroup behaviour after the establishment of the groups. In particular, the establishment of groups has a cooperative element since children mutually have to agree on these. This collective agreement might diminish the effect of a previously elicited competitive orientation and consequently not affect intergroup behaviour. The mutual agreement might not have the same salience when groups have been established before allowing the competitive orientation to shape the perception of the relation of groups. This interpretation might be an interesting avenue for future research. One could investigate whether a previous group manipulation is a necessary condition for the effects of cooperation and competition found by Spielman [31]." (p. 11)

Comment 2:

The authors seem to call into question the presence of minimal group effects on sharing based on their null effect. However, no manipulation check was conducted to demonstrate that the minimal group manipulation was successful. It is possible that children did indeed care about their minimal groups yet this had no effect on their sharing. However, it is also possible that the minimal group manipulation did not successfully induce strong ingroup preferences. If the authors had conducted a complementary preference check, perhaps at the very end of the task, their claims about the lack of a group effects on sharing would be much better-grounded. The fact that they saw a group effect in the inclusion task is somewhat helpful in this regard. However, in the absence of a manipulation check, it is difficult to know how strong ingroup preferences were (e.g., maybe a very weak ingroup preference is enough for the inclusion task but not the sharing task) and/or whether some children showed an ingroup preference while others did not which could help account for the sharing results.

Response:

We thank Reviewer 2 for this comment. In our study, we do not find any sign for children's in-group bias. To ensure that children understood the minimal group situation, we conducted a comprehension check and children had to state their own group membership and whether they shared group membership with the portrayed children (for the dictator game) and the puppets (for the inclusion task). For the dictator game, three children did not pass this comprehension check and were excluded from the respective analysis. For the inclusion task, all children passed the comprehension check. Thus, all data included in the statistical analyses are from children who understood that there were in different groups and could identify their own and their interactants group membership. Accordingly, the necessary conditions for the detection of an in-group bias are fulfilled. Also, our procedure was very similar (or identical) to previous work which detected in-group biases in children (e.g., Toppe et al., 2020; Dunham et al., 2011).

Given that evidence on preschool-aged children's in-group bias is somewhat mixed and that in-group bias seems to develop throughout preschool age (Over, 2018), our null-result seems plausible.

However, we fully acknowledge that our evidence is not conclusive in this point. Particularly, in early childhood research were minor changes in procedures can affect results substantially (e.g., for minimal group manipulations, see Misch & Dunham, 2021). Therefore, we included the argument in the Discussion. The critical passage reads as follows:

“Interestingly, we did not find an impact of group membership on children’s sharing and no developmental trend in children’s social inclusion behaviour. In other words, there is no indication of an in-group bias in our data. This finding is in contrast to evidence suggesting the emergence of an in-group bias in minimal group contexts around preschool age from similar cultural contexts (e.g., [17,19,23,25,26,94]). Our results are in line with the studies by Spielman [31] and Plötner et al. [24], who found children’s sharing to be independent of the recipients’ group membership, suggesting that children’s sensitivity for conventional groups seems to emerge after preschool age. Notably, all children who were part of the statistical analyses passed a comprehension check and could identify their own and the interactants’ group memberships correctly. Only three children did not pass the comprehension check and were excluded from data analyses. Children appear to have a robust capacity to perceive such groups, but the effect of such groups on their prosocial behaviour seems rather fragile. However, as children’s prosocial behaviours are poorly related [104,105], one should be careful when claiming that children do not generally show an in-group bias in minimal group contexts. It might be that children’s in-group bias is expressed by helping or affiliative behaviours (e.g., liking), which we did not assess in the current study. Also, it might be that our measures were not sensitive enough to detect the effect of the groups. Nevertheless, our null results for children’s in-group bias point to the value of studies examining the effects of minimal groups on diverse prosocial behaviours throughout early and middle childhood. Ideally, such an investigation is organized in a collaborative research project involving multiple laboratories conducting the same procedure and thereby generating reliable data (similar to [106–109]). Further, it might be interesting to take a closer look to procedural details (e.g., how exactly is group membership established) to learn more about the effects found in previous studies. Open materials seem a promising solution in this endeavour.” (p. 11-12)

Dunham, Baron, & Carey (2011). Consequences of 'Minimal' Group Affiliations in Children. *Child Development*, 82(3), 793–811.
Misch & Dunham (2021). (Peer) Group Influence on Children's Prosocial and Antisocial Behavior. *Journal of Experimental Child Psychology*, 201: 104994.

- Misch & Dunham (2021). (Peer) Group Influence on Children's Prosocial and Antisocial Behavior. *Journal of Experimental Child Psychology*, 201: 104994.
- Over (2018). The Influence of Group Membership on Young Children's Prosocial Behaviour. *Current Opinion in Psychology*, Early Development of prosocial behavior, 20: 17–20.
- Toppe, Hardecker, & Haun (2020). Social Inclusion Increases over Early Childhood and Is Influenced by Others' Group Membership. *Developmental Psychology*, 56(2), 324–35.

Comment 3:

As mentioned above, it is great that this study was preregistered. However, there were a few differences between the preregistration and the study that gave me pause. First, the pre-registration places a lot of emphasis on questions surrounding Social Dominance Orientation (SDO). However, this is conspicuously absent from the MS. For instance, more than half of the section under “Specify exactly which analyses you will conduct to examine the main question/hypothesis” in the preregistration is devoted to SDO yet this is not mentioned at all in the MS. It seems important, at the very least, to address this discontinuity. Second, the pre-registration states that 192 children will be tested but the MS reports that N = 144 were tested. Later in the MS, it is clear that 96 children were tested in the solitary condition which would seem to indicate that the N was indeed 192 (48 in cooperative, 48 in competitive, 96 in solitary) which means that there may simply be an error under “design and procedure” but it left me a bit confused as to how many children were actually included. If there is indeed a discrepancy between the MS and the prereg, this should be addressed explicitly. Particularly given that the reported power analysis is associated with an N of 144.

Response:

We acknowledge Reviewer 2's comment and revised to manuscript at different parts. To clarify: 192 children participated in the study but since 48 children in the solitary condition were observers and their data was not included into the analyses, we only analysed data from 144 children. One critical part of the Materials and Methods now reads as:

“We randomly assigned dyads to one of three experimental conditions: Cooperative (n = 48 children), competitive (n = 48 children), and solitary (n = 96 children). Importantly, we tested twice as many children in the solitary condition (n = 96 children) since only one child interacted with the gaming apparatus (targets; n = 48 children) while the other child was parallelly engaging in a non-gaming activity (observers; i.e., drawing a picture; n = 48 children; details see below).

In deviation from our preregistration, we only analysed the data of those children in the solitary condition who played the intervention game (targets; n = 48 children) and excluded the data of children who did not play the intervention game (observers; n = 48 children) from our inferential analyses. Our initial idea was to analyse the data of the observers as well since these might constitute a non-gaming baseline. However, most observers oriented toward the intervention game frequently and were slightly frustrated that their partner (but not they) could play the game. From our view, this situation does not constitute a baseline. Besides, we did not have any prediction on how this particular social comparison affects children's subsequent prosociality. Thus, we decided to exclude these children from the data analysis but report the descriptive results of this subsample. Importantly, already in the conception of the study, the observer sample was planned as an additional sample. That is, our prior power analysis (which determined the sample size of the three central conditions) was not planned with the observer condition.” (p. 6)

Also, we included the analyses on the effect of parents' social dominance orientation into to the manuscript. This changed several parts of the manuscript. One critical part of the Introduction reads as follows:

“Finally, we aimed to learn more about the impact of children's socialisation context on their in-group bias. To contribute to this line of research, we surveyed children's parents and assessed their social dominance orientation (SDO; [60]). SDO is a promising proxy for parents' intergroup socialisation of their children, being one of the strongest predictors of parents' own intergroup attitudes and behaviour [61]. SDO is assumed to be a stable trait indicating the preference for inequality among social groups [60,61]. People scoring high on SDO tend to perceive the world as competitive, leading them to feelings of dominance and superiority [62]. In adults, SDO is strongly linked to diverse intergroup behaviours [60,61,63]. Amongst others, SDO is related to prejudice toward familiar groups and affiliation toward minimal groups, with adults scoring higher on SDO showing a stronger in-group

preference (e.g., [60,64,65]). Furthermore, adults who score high on SDO tend to approve of unequal resource distributions [66].

For the link between parental intergroup attitudes and children's social behaviour, mixed findings exist, with studies indicating positive [67–70], negative [68], and no connections [68,71,72]. In their meta-analysis, Degner & Dalege [73] found a moderate positive correlation ($r = .38$) between parents and their children's intergroup attitudes. Previous studies also found significant associations between adolescent children's and their parents' SDO [74,75], whereas only a weak association was found for 8- to 10-year-old children [76]. Further, 4- to 5-year-olds were more likely to punish an in-group member who acted unfairly in an intergroup context when their parents' SDO was lower, indicating an increased sense of fairness in children whose parents score low on SDO [77]. These findings suggest that parental SDO appears to be related to their preschool-aged children's intergroup behaviour. Thus, in addition to Spielman [31] and Toppe et al. [56], we assessed parents' SDO to predict their children's sharing and social inclusion behaviour in an intergroup context to learn more about the impact of children's socialisation contexts on these behaviours.” (p. 4)

Comment 4:

On the subject of the power analysis, it would be helpful to make clear what effect was being tested in the power analysis. Given that the interaction between age and condition was a focus of the MS and preregistration, was this the effect of interest in power analyses? Or was the study powered to detect a main effect of condition? If not the age x condition interaction, this would need to be addressed by, minimally, labelling these analyses as exploratory. Additionally, how were power analyses run? Were simulations used to accommodate the mixed models, was power approximated using something like g*power or something else? More details here would be very helpful.

Response:

We thank Reviewer 2 for this comment. The power analysis referred to the overall effect of the predictor variables (i.e., the interaction of age and condition and main effects of condition, age, and engagement). We added this information to the methods section:

“The sample size of $N = 144$ was suggested by a prior power analysis expecting a medium effect with a statistical power of .80 and a probability of .05 for a type I error. This power analysis referred to the overall effect of the variables of interest—interaction between condition and age and the main effects of condition, age, and engagement. The power analysis was conducted with the pwr package [83] in R statistical software ([84]; see R script at osf.io/pu89t/).” (p. 5)

Comment 5:

Finally, given that this paper focuses on children, I had several questions about the way age was dealt with. As mentioned above, it was unclear to me if the interaction between age x condition was the main effect of interest. If so, I was curious about how children were recruited to ensure roughly even spread across age. Or, if recruitment was not directed in this way, what was the spread across ages that organically emerged from your sampling and how might this influence age-based analyses?

Response:

In the revised manuscript, we give a written breakdown of the sample for each age group. Further, we offer a histogram of children's age in the supplemental material. The Participants section now reads:

“The sample used for analysis consisted of 144 German children aged between 4 and 6 years (50% female; mean age = 4.96 years; age range = 4.03 to 6.05 years). Children were from a mid-sized German city, and recruitment was based on a laboratory-maintained database, including children from about 150 day-care centres. Participants tested in this study were from 20 day-care centres located in different districts of the city, allowing the assumption that children had diverse socio-economic backgrounds. We aimed to include age as a continuous variable into the statistical analysis and, thus, wanted children's age to be distributed evenly. To achieve a relative even distribution of age across conditions, about one half of the sample should be aged between 4 and 5 and the other half aged between 5 and 6 within each condition ($n_{4y/o} = 78$, $n_{5y/o} = 63$, $n_{6y/o} = 3$; for a histogram of children's age, see supplementary material figure S1).” (p. 5)

Also, we thank Reviewer 2 for pointing to unclarity regarding the interaction between children's age and the experimental condition. We specified this aspect in the manuscript. The critical part now reads as:

"For all hypotheses, we investigated in how far children's age accentuates the effect of the experimental conditions by considering the interaction between condition and age. The consideration of this interaction term seems highly relevant as the understanding of cooperative and competitive contexts seems to emerge throughout preschool age [78–82]." (p. 4)

Comment 6:

I found the introduction and, to a lesser extent the discussion, quite hard to navigate because it jumps around a fair bit in terms of content and organization. The discussion was slightly easier to follow thanks to the subheadings, but I would urge the authors to run through both sections again, making sure that the content is organized in the most streamlined way and smoothing out the grammar (e.g., verb tenses jump around a bit too).

Response:

We think this is a good suggestion and revised the Introduction and the Discussion. Also, we added subheadings to the Introduction.

Comment 7:

I had to reread the procedure for the inclusion task several times before understanding the difference between the 'normal' rounds and the directive trial. Specifically, that in the former the puppets spoke out loud but *to themselves* while in the latter they spoke to the child. I now understand this distinction but it might still be worth highlighting this difference very clearly so that it's apparent how the directive trial is different.

Response:

We revised the wording of the procedure to make it more comprehensible. Also, we uploaded a demonstration video of the task on the Open Science Framework (osf.io/pu89t/). The revised passage reads as follows:

"Hereafter, the in-group puppet introduced a ball-tossing game and revealed the covered apparatus. The in-group puppet and the child passed the ball back and forth through each of the three tubes of the apparatus. The in-group puppet stayed at one corner of the apparatus (counterbalanced) and initiated another two rallies. When the in-group puppet held the ball, the out-group puppet appeared at the vacant corner of the triangle stating "Hello". While holding the ball, the in-group puppet decided to pass the ball to the child after speaking with itself and thinking aloud about to whom it would pass the ball to ("Do I pass the ball to Name of out-group puppet or to Name of child?"). Children could freely decide to which of the puppets they pass the ball. Both puppets always passed the ball to the child. If not included for two consecutive rallies, the out-group puppet gave a prompt indicating the desire to be included when the in-group puppet held the ball ("Can I join your game?"). Again, the in-group puppet decided to pass the ball to the child after weighing both alternatives and thinking aloud.

Four rallies were played in this way, followed by a final directive trial, in which the in-group puppet addressed the child directly and asked to whom it should pass the ball (to the out-group puppet versus the participant; "To whom should I pass the ball? To you or to Name of out-group puppet?").

When children were holding the ball, the experimenter avoided eye-contact and faced the floor. If children did not pass the ball for about 10 seconds, the in-group puppet encouraged them to pass the ball ("Now, it is your turn."). A demonstration video of this task is available at osf.io/pu89t/." (p. 8)

Comment 8:

I understand the authors' rationale for not reporting data for the 'baseline' children and agree that this condition does not seem like a clear baseline. However, it would still be worth reporting their data in the interest of transparency. Their post-hoc exclusion made me wonder whether their data were hard to explain for some reason.

Response:

Agreed. In order to make our data more transparent, we report the descriptive results of the 'observer' subsample for each dependent variable (expect children's engagement). However, we did not include this subsample into our model analyses as the 'observer' role cannot be considered as a baseline and is hard to compare with the other three conditions. The descriptive results of the 'observers' can be found in the tables of the Results section.

Comment 9:

It would be helpful if the authors could be clear about when they are referring to dyads vs participants (e.g., N = 48 participants, N = 24 dyads). I think this might help clear up some general confusion about the sample size.

Response:

We thank Reviewer 2 for pointing to this unclarity. We changed the respective parts and hope that the manuscript is clearer in this regard.

Comment 10:

Since the authors already employ a partial Bayesian approach and are thus familiar with Bayesian methods, I wondered if they considered running their models in a Bayesian framework. Of course, I appreciate that this was not the preregistered approach and certainly wouldn't advocate for replacing the current models. However, since the authors make claims that hinge on null findings, a Bayesian approach would allow them to assign a probability to the null hypothesis.

Response:

We think that this is a good suggestion and added Bayesian factors to the statistical analysis. To calculate Bayes factors, we used the bayestestR package. The critical part of the manuscript reads as:

"Further, we calculated Bayes factors (BFs) comparing the evidence for the full model with the evidence for the null model given our data. Bayes factors are a numerical value quantifying how well a hypothesis predicts the given data relative to a competing hypothesis [98]. Here, we used the bayestestR package [99] to extract the Bayes factor for the full and the null model and applied Jeffrey's system [100] to interpret these. The advantage of Bayes factors is that they allow for a differentiated interpretation of the support of the null versus the alternative hypothesis." (p. 9)

These Bayesian factors strongly support the null hypotheses for our findings related to children's sharing and social inclusion (see Results).